# SCALABLE BAYESIAN LEARNING WITH POSTERIORS

**Samuel Duffield**[*]
Normal Computing

**Kaelan Donatella**
Normal Computing

**Johnathan Chiu**
Normal Computing

**Phoebe Klett**
Normal Computing

**Daniel Simpson**
Normal Computing

## ABSTRACT

Although theoretically compelling, Bayesian learning with modern machine learning models is computationally challenging since it requires approximating a high dimensional posterior distribution. In this work, we (i) introduce posteriors, an easily extensible PyTorch library hosting general-purpose implementations making Bayesian learning accessible and scalable to large data and parameter regimes; (ii) present a tempered framing of stochastic gradient Markov chain Monte Carlo, as implemented in posteriors, that transitions seamlessly into optimization and unveils a minor modification to deep ensembles to ensure they are asymptotically unbiased for the Bayesian posterior, and (iii) demonstrate and compare the utility of Bayesian approximations through experiments including an investigation into the cold posterior effect and applications with large language models.

posteriors: github.com/normal-computing/posteriors

## 1 INTRODUCTION

Bayesian learning is a framework for updating and inferring unknown model parameters in the presence of data. It is based on Bayes' theorem:

$$p(\theta \mid y_{1:N}) \propto \underbrace{p(\theta)p(y_{1:N} \mid \theta)}_{\text{offline}} \propto \underbrace{p(\theta \mid y_{1:N-1})p(y_N \mid \theta)}_{\text{online}}, \tag{1}$$

which provides a principled way to update our beliefs about the parameters $\theta \in \mathbf{R}^d$ as data is observed once $y_{1:N}$ or sequentially $y_N$. Bayesian learning represents uncertainty about the parameters using the posterior distribution $p(\theta \mid y_{1:N})$, which combines prior beliefs $p(\theta)$ with the likelihood or generating process $p(y \mid \theta)$ for the data. This contrasts with optimization approaches, which store a single point estimate of the parameters $\theta^* = \arg\max_\theta p(y_{1:N} \mid \theta)$. In our view, for modern machine learning models and pipelines, Bayesian learning offers three key advantages, which are visualized in Figure 1: (a) improved generalization and out-of-distribution predictions, (b) coherent online learning of new information (without catastrophic forgetting), and (c) the ability to decompose predictive uncertainty into aleatoric (natural or explained uncertainty in the data) and epistemic (unexplained uncertainty that diminishes with more training data) components. It's important to note that in the more classical setting of statistical models with interpretable parameters, there are additional benefits to be gained from Bayesian learning - these include propagation of uncertainty (Duffield et al., 2024) and model validation (Gelman et al., 2020). In this work, we introduce posteriors, a python package designed to be minibatch-first and easily applicable to large-scale models with PyTorch (Paszke et al., 2019) and Hugging Face (Wolf et al., 2020) whilst also being compatible with classical Bayesian models through composition with Pyro (Bingham et al. 2019, Appendix G).

Generalization is a key metric for machine learning models, quantifying the ability of the model to perform tasks beyond its training set. Bayesian predictions offer a compelling approach to mitigate overfitting by averaging over a posterior distribution of plausible models rather than relying on a single fit. In various settings, Bayesian methods have demonstrated improved out-of-distribution performance (Neal, 2012; Blundell et al., 2015; Liu et al., 2020).

---

[*]Correspondence to: sam@normalcomputing.ai

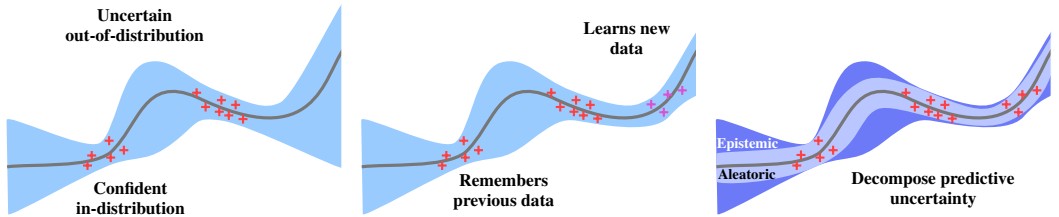

Figure 1: **Pictorial representation of the benefits of Bayesian learning.** Left: Averaging over multiple plausible fits to the data improves out-of-distribution generalisation. Center: Adapting to new online data without forgetting previous data. Right: Decomposing predictive uncertainty, with epistemic uncertainty providing an improved indicator for out-of-distribution detection.

Another vital consideration for modern machine learning pipelines is that of online or continual learning, where the model continuously updates and learns from new data in real-time without needing to be retrained from scratch. Bayes' theorem (equation 1) provides a coherent framework for online learning, where new data can be sequentially incorporated into the model by using the posterior distribution as the prior on receipt of the next data batch. When implemented exactly, this is equivalent to learning all data $y_{1:N}$ at once equation 1, and all data points are exchangeable and treated identically. This contrasts to naive online optimization, which typically results in the catastrophic forgetting (Goodfellow et al., 2013; Kirkpatrick et al., 2017) of previously learnt tasks.

In addition, it is essential for statistical models to be auditable and reliable, especially when they are met with data that is far outside the training distribution. With an optimization approach, the model is only able to provide predictions in the form of total uncertainty for the predictive distribution over test inputs $x^*$, via the entropy $H[p(y \mid x, \theta^*)] := -\mathbb{E}_{p(y|x,\theta^*)}[\log p(y \mid x, \theta^*)]$. This is in contrast to Bayesian methods, which can decompose the total uncertainty (TU) of the posterior predictive distribution $p(y \mid x) = \mathbb{E}_{p(\theta|y_{1:N})}[p(y \mid x, \theta)]$ into two components (Wimmer et al., 2023; Sale et al., 2023; Hofman et al., 2024):

$$\text{TU} := H[p(y \mid x)], \quad \text{AU} := \mathbb{E}_{p(\theta|y_{1:N})}[H[p(y \mid x, \theta)]], \quad \text{EU} := \text{TU} - \text{AU}. \tag{2}$$

Aleatoric uncertainty (AU) represents natural uncertainty in the data, such as synonyms or sentence starts in natural language generation. Epistemic uncertainty (EU) represents uncertainty in the model parameters and diminishes with more data (as $p(\theta \mid y_{1:N}) \to \delta(\theta \mid \theta^*)$ as $N \to \infty$) and therefore is a better indication of model uncertainty in predictions. This decomposition is crucial for understanding model behaviour and making informed decisions in high-stakes applications (Kendall and Gal, 2017), although is not unique to Bayesian models but rather the broader concept of second-order uncertainty (Osband et al., 2024; Sale et al., 2023; Hofman et al., 2024).

In this work, we concisely survey techniques for Bayesian learning with a view to scalability in both large data and large parameter regimes (with an extensive survey found in Appendix A). We then make the following contributions:

1. We introduce `posteriors`; a functional, minibatch-first PyTorch (Paszke et al., 2019) package for scalable and extensible Bayesian learning.

2. We formalize gradient descent methods as the low-temperature limit of stochastic gradient Markov chain Monte Carlo (SGMCMC) methods, thus implying a simple modification of deep ensembles to parallel SGMCMC that ensures convergence to the Bayesian posterior.

3. We provide experiments demonstrating the broad utility of `posteriors` and the accessibility of the aforementioned benefits. We investigate the generalization performance of a host of approximate Bayesian methods and the prominence of the *cold posterior effect* (Wenzel et al., 2020; Izmailov et al., 2021), composing `posteriors` with LoRA (Hu et al., 2021) to mitigate forgetting in a continual learning task with Llama 2 (Touvron et al., 2023) and out-of-distribution detection via epistemic uncertainty with Llama 3 (AI@Meta, 2024).

We conclude with a discussion of related work and an outlook on future avenues for research and impact.

## 2 BAYESIAN LEARNING

We now review the most prominent Bayesian learning methods for large-scale problems, all of which are implemented in `posteriors`. An in-depth survey with full details and discussions on caveats and computational bottlenecks can be found in Appendix A.

A **Laplace approximation** is motivated by a second-order Taylor expansion around the mode of the posterior distribution $\theta_{\text{MAP}} = \arg\max_\theta p(\theta \mid y_{1:N})$. This gives a Gaussian approximation $p(\theta \mid y_{1:N}) \approx \mathbf{N}(\theta \mid \theta_{\text{MAP}}, \hat{\Sigma})$. In the supervised learning setting, the covariance $\hat{\Sigma}$ is typically chosen (Daxberger et al., 2021) to be an approximation to the Fisher information matrix $\mathrm{F}(\theta) = \mathbb{E}_{p(x,y|\theta)}[\nabla_\theta^2 \log p(\theta \mid x, y)]$ which is intractable as we do not have access to the true distribution over inputs $p(x)$. Instead, two approximations (Kunstner et al., 2019) are used based on the training data $\hat{p}(x, y) = N^{-1} \sum_{i=1}^N \delta(x, y \mid x^i, y^i)$. The first we denote the *empirical Fisher* $\mathrm{F}_{\text{E}}(\theta) = \mathbb{E}_{\hat{p}(x,y)}[\nabla_\theta \log p(\theta \mid y) \nabla_\theta \log p(\theta \mid y)^\top]$ and is easy to compute. The second we denote the *conditional Fisher* $\mathrm{F}_{\text{C}}(\theta) = \mathbb{E}_{\hat{p}(x)p(y|x,\theta)}[\nabla_\theta \log p(\theta \mid y) \nabla_\theta \log p(\theta \mid y)^\top]$ and can be computed efficiently for common neural network likelihoods via an equivalence with the Generalized Gauss-Newton matrix (GGN, Martens 2020) with further details in Appendix A.1.

In its most common form **variational inference** (VI, Blei et al. 2017) also forms a Gaussian approximation $p(\theta \mid y_{1:N}) \approx q_\phi(\theta) = \mathbf{N}(\theta \mid \mu, \Sigma)$. However, VI directly learns the mean and covariance $\phi = (\mu, \Sigma)$ through an optimization routine to minimize the Kullback-Leibler objective $\mathrm{KL}[q_\phi(\theta) \mid\mid p(\theta \mid y_{1:N})]$, whose gradient can be approximated with Monte Carlo samples and tricks including reparameterzation (Rezende et al., 2014) and stick-the-landing (Roeder et al., 2017).

An important caveat to note for Gaussian approximations is that the full covariance matrix requires $\mathcal{O}(d^2)$ memory and often $\mathcal{O}(d^3)$ runtime for the required inversion and sampling operations. In the large parameter regime, this is prohibitive, and further approximations are required (MacKay, 1992; Ritter et al., 2018; Daxberger et al., 2021). Additionally, we can reduce memory at inference time by constructing uncertainty directly in function space by **linearizing the forward model** (Foong et al., 2019; Immer et al., 2021). If the likelihood has the form $p(y \mid x, \theta) = p(y \mid f_\theta(x))$ and we have a posterior approximation $\mathbf{N}(\theta \mid \mu, \Sigma)$ then the linearized forward distribution becomes $\mathbf{N}(f(x) \mid f_\mu(x), \nabla_\theta f_\mu(x)^\top \Sigma \nabla_\theta f_\mu(x))$ where $\nabla_\theta f_\mu(x)$ is the Jacobian of $f_\theta(x)$ evaluated at $\mu$. This avoids having to sample many large parameter vectors.

**Stochastic gradient Markov chain Monte Carlo** (SGMCMC, Ma et al. 2015; Nemeth and Fearnhead 2021) instead forms a Monte Carlo approximation $p(\theta \mid y_{1:N}) \approx N^{-1} \sum_{k=1}^K \delta(\theta \mid \theta_k)$. The samples $\{\theta_k\}_{k=1}^K$ are typically collected by (approximately) evolving a stochastic differential equation (SDE) - see Section 4 and Appendix B - and collecting samples along the trajectory. Traditional full-batch Markov chain Monte Carlo (MCMC) schemes (Andrieu et al., 2003) use a Metropolis-Hastings step to ensure convergence of the Monte Carlo approximation is preserved through discretization. In the minibatch setting, the Metropolis-Hastings ratio cannot be estimated easily. Instead, suppose we have access to an unbiased stochastic gradient. Then, as noted in Welling and Teh (2011), the bias in the trajectory decreases quadratically $\mathcal{O}(\epsilon_t^2)$ for learning rate $\epsilon_t$. Therefore, adopting a suitably decreasing learning rate schedule $\sum \epsilon_t^2 < \sum \epsilon_t = \infty$ (Robbins and Monro, 1951) will ensure an asymptotically unbiased Monte Carlo approximation.

## 3 POSTERIORS: UNCERTAINTY QUANTIFICATION WITH PYTORCH

`posteriors` is designed to be a comprehensive library for uncertainty quantification in deep learning models. The key features outlining the `posteriors` philosophy are:

- **Composability**: `posteriors` is written in PyTorch (Paszke et al., 2019) and has compatibility with a broad range of tools including the Llama (Touvron et al., 2023) and Mistral (Jiang et al., 2023) models, `lightning` (Falcon and The PyTorch Lightning team, 2019) for convenient logging and device management, optimizers from `torchopt` (Ren et al., 2023) and probabilistic programs from `pyro` (Bingham et al., 2019), see Appendix G.

- **Extensible**: The transform framework (Figure 3) adopted by `posteriors` is very general and allows for the easy adoption of new algorithms. Additionally, `posteriors` supports

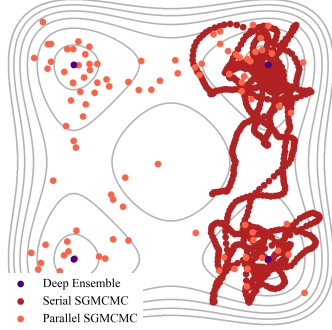

Figure 2: **Trajectories of various sampling methods** for a toy multimodal posterior. Deep ensemble concentrates on the modes, serial SGMCMC struggles to transfer between modes, parallel SGMCMC combines the benefits of both.

```
1   # Load classifier model
2   # Load dataloader
3   num_data = len(dataloader.dataset)
4
5 ∨ def log_posterior(params, batch):
6       inputs, labels = batch
7       logits = torch.func.functional_call(model, params, inputs)
8 ∨     log_post_val = (
9           -torch.nn.funtional.cross_entropy(logits, labels)
10          + posteriors.diag_normal_log_prob(params) / num_data
11      )
12      return log_post_val, logits  # Return value and auxiliary info
13
14
15 ∨ transform = posteriors.vi.diag.build(
16      log_posterior, torchopt.adam(), temperature=1/num_data
17  )   # Can swap out for any posteriors algorithm
18
19  state= transform.init(params)
20
21 ∨ for batch in dataloader:
22      state, aux = transform.update(state, batch)
```

Figure 3: `posteriors` **code snippet** to train a classifier with variational inference. `posteriors` recommends normalising the log posterior across the batch with scale independent of batch size or $N$. Scaling the prior and temperature by $N^{-1}$ ensures the posterior is still correctly targeted.

arbitrary likelihoods rather than being restricted to hard-coded regression or classification as is common in other libraries (Daxberger et al., 2021; Detommaso et al., 2023).

- **Functional**: `posteriors` adopts a functional API via PyTorch's (Paszke et al., 2019) functional module. As championed by the JAX (Bradbury et al., 2018) ecosystem, the functional approach makes for code that is easier to test and compose with other functions. Importantly for `posteriors`, functional programming is also closer to the mathematical description, which is particularly useful for Bayesian modelling.

- **Scalable**: Unlike most common Bayesian software (Carpenter et al., 2017; Cabezas et al., 2024; Bingham et al., 2019), `posteriors` is **minibatch-first**, thus allowing for efficient computation in large datasets. Additionally, flexible sub-spacing is provided for scaling to large models.

- **Swappable**: The transform framework (Figure 3) allows users to switch between approaches seamlessly, allowing them to experiment and find the best method for the use case.

This combination of features makes `posteriors` unique amongst the PyTorch ecosystem which hosts the majority of the popular pretrained Hugging Face models (Wolf et al., 2020).

### 3.1 SUPPORTED METHODS AND COMPLEXITIES

All methods discussed in Section 2 are included in `posteriors`. Flexible Laplace approximations are supported based on both the empirical Fisher and conditional Fisher (via the GGN matrix, see Section 2 and Appendix A). Laplace approximations rely on an optimization routine to find the MAP followed by a $O(bd)$ time and memory operation (for batch size $b$) to construct diagonal Fisher information matrices which ultimately require $O(d)$ memory. For very large models this can be a memory bottleneck meaning the batchsize may need to be reduced for the Laplace construction in comparison to the optimization (although the Laplace construction requires only a single epoch).

Variational inference for fitting a Gaussian distribution with diagonal covariance costs $O(kd)$ time and memory per iteration where $k$ is the number of samples from the variational distribution which is often set to 1 although larger $k$ reduces the variances of the gradient estimation. Dense covariance versions of both Laplace and VI are also supported but require $O(d^2)$ memory and $O(d^3)$ for sampling, however, this can still be useful for machine learning models when combined with subspace approaches (Mangrulkar et al., 2022; Daxberger et al., 2021).

As we will see in Section 4, SGMCMC is closely related to stochastic gradient descent and has the same per-iteration complexity. Although, a key difference is the final output is an ensemble of

parameters rather than the single point estimate. As such memory requirements are $O(Md)$ where $M$ is the number of output samples, as with deep ensembles (Lakshminarayanan et al., 2017). Further, as a suitable decay in autocorrelation is desired between samples the number of iterations is typically longer for SGMCMC than optimization, although this can be parallelized, see Section 4.2.

In addition, `posteriors` hosts extended Kalman methods (Ollivier, 2018) (which has the same complexity as VI, Zhang et al. 2018), unified optimization via torchopt (Ren et al., 2023) and has a roadmap of methods to be added within the general framework including SVGD (Liu and Wang, 2016) and more advanced SGMCMC (Ma et al., 2015; Zhang et al., 2020). There is also a host of general-purpose utility functions not found elsewhere in the PyTorch ecosystem. This includes linearized forward calls, Fisher/GGN matrices, their diagonal counterparts, fast Fisher/GGN vector products and the conjugate gradient method, which provides the tools to implement Hessian-free optimization (Martens et al., 2010) and experiment with other second-order methods (Titsias, 2024; Duffield and Singh, 2022).

## 3.2 LIMITATIONS

`posteriors` supports arbitrary likelihoods and loss functions; as a result methods that utilise the specific form of the loss function (normally as classification or regression), e.g. the deterministic last layer approach in Harrison et al. (2024) or the probit approximations from Daxberger et al. (2021), are currently out of scope.

Similarly, the current methods in `posteriors` do not assume any specific structure of the model or its parameters, see Figure 3. However, this makes it more challenging to support model-aware techniques such as DVI (Wu et al., 2018) or K-FAC (Martens and Grosse, 2015), although the benefits of such approaches may warrant an inclusion in the library going forward. Specifically for K-FAC, the Fisher information matrix is assumed to exhibit a local block structure that admits cheap inversion and sampling via Kronecker-factors. Initially, K-FAC was only developed for feedforward layers (Martens and Grosse, 2015), before being extended to convolutional (Grosse and Martens, 2016) and recurrent (Martens et al., 2018) networks. More recently, it was generalized to weight-sharing networks (Eschenhagen et al., 2024) (including transformers and graph neural networks). Developing a flexible K-FAC submodule within `posteriors`' functional framework is a high priority. Such a module should be suitably model-agnostic and capable of being applied across various inference methods (e.g., optimization, Laplace, VI). This advancement would enable scalable inference with non-diagonal Fisher information matrices, broadening the library's capabilities.

Convergence diagnostics (e.g. Cowles and Carlin 1996) represent a key component of MCMC workflow, however are not included directly in `posteriors` due to the easy composition with Pyro's (Bingham et al., 2019) extensive suite of convergence diagnostics. Similarly, `posteriors` is an inference library rather than a probabilistic programming language, therefore does not include tools such as distribution definitions which can be utilized through composition with Pyro as demonstrated in Appendix G. Finally, we opt not to include a `fit` convenience function (Chollet et al., 2015; Daxberger et al., 2021) in favour of the more flexible and transparent `update` framework in Figure 3.

## 4 STOCHASTIC GRADIENT DESCENT, SGMCMC AND DEEP ENSEMBLES

This section describes a tempered version of SGMCMC that introduces an additional temperature parameter that allows us to view SGMCMC as a generalization of stochastic gradient descent (SGD) and variants. This viewpoint has guided the implementation of `posteriors` allowing users access to SGMCMC within a familiar functional framework of popular optimization libraries such as torchopt (Ren et al., 2023) and optax (DeepMind et al., 2020), thus making Bayesian learning scalable and maximally accessible to the broader machine learning community. In the second half of this section, we show how this points to a small modification of deep ensembles (Lakshminarayanan et al., 2017) to ensure convergence to the posterior, and therefore how `posteriors` can be used to easily convert a deep ensemble pipeline into one of (parallel) Bayesian learning. Comprehensive details on SGMCMC are found in Appendix A.3.

The tempered connection between SGMCMC and SGD has been discussed in previous works for specific optimizers/samplers e.g. Wenzel et al. (2020), however here we give a complete characterization of the connection based on the framework in Ma et al. (2015) which dictates the implementations in

`posteriors`. Additionally, to our knowledge, the corollary unifying deep ensembles and parallel SGMCMC is novel.

The minibatch-first nature of `posteriors` alongside the reframing of stochastic gradient optimization as SGMCMC results in an implementation with access to both approaches within a unified syntax as well as enabling the codesign of new algorithms, all within a familiar optimization-style framework.

## 4.1 A UNIFIED FRAMEWORK FOR SGD AND SGMCMC

Consider targeting a tempered target distribution $\pi_{\mathcal{T}}(z) \propto \pi(z)^{\frac{1}{N\mathcal{T}}}$ for temperature $\mathcal{T} \in [0, \infty)$, which can be considered as a synthetic hyperparameter controlling the variance of $\pi_{\mathcal{T}}(z)$ (Wenzel et al., 2020; Ding et al., 2014). Here $z = (\theta, \omega)$ where $\omega$ are any auxiliary parameters such as momenta (Chen et al., 2014) or thermostat (Ding et al., 2014). The distribution $\pi(z)$ is thus an extended target with $p(\theta \mid y_{1:N})$ marginal in $\theta$.

An SDE is invariant for $\pi_{\mathcal{T}}(z)$ if and only if it has the following form (Ma et al., 2015)

$$dz = \mathcal{T}^{-1}[\mathrm{D}(z) + \mathrm{Q}(z)]N^{-1}\nabla \log \pi(z)dt + \nabla \cdot [\mathrm{D}(z) + \mathrm{Q}(z)]dt + \sqrt{2\,\mathrm{D}(z)}dw,$$

$$\implies dz = [\mathrm{D}(z) + \mathrm{Q}(z)]N^{-1}\nabla \log \pi(z)dt + \mathcal{T}\,\nabla \cdot [\mathrm{D}(z) + \mathrm{Q}(z)]dt + \sqrt{2\,\mathcal{T}\,\mathrm{D}(z)}dw, \quad (3)$$

where the second line comes from rescaling $\mathrm{D}(z), \mathrm{Q}(z) \to \mathcal{T}\,\mathrm{D}(z), \mathcal{T}\,\mathrm{Q}(z)$ [1].

Setting $\mathcal{T} = N^{-1}$ regains the original target, but conveniently, the scale of the drift term $N^{-1}\nabla \log \pi(z)$ no longer grows with $N$ as $\log \pi(\theta) = \log p(\theta) + \sum_{n=1}^{N} \log p(y_n \mid \theta) + \text{const}$. This normalization of the loss or log posterior is the convention for optimization and results in more consistent learning rate selection across problems and is recommended in `posteriors` - see Figure 3.

Intuitively, the low-temperature limit $\mathcal{T} \to 0$ converts the target distribution $\pi(z)$ into one with all probability mass concentrated on (local) maxima and transparently the SDE equation 3 into an ODE

$$\mathcal{T} \to 0 \implies dz = [\mathrm{D}(z) + \mathrm{Q}(z)]N^{-1}\nabla \log \pi(z)dt, \quad (4)$$

which represents exactly a variant of continuous time gradient ascent on $N^{-1} \log \pi(z)$.

All SGMCMC algorithms in `posteriors` feature the temperature $\mathcal{T}$ as a parameter with $\mathcal{T} = 0$ resulting in a valid SGD variant (specific examples can be found in Appendix C). The extensibility and strict support for minibatches allow a unified `posteriors`' implementation and new procedures immediately apply to both SGMCMC and SGD (e.g. step-size adaptation, which has in classical Bayesian statistics utilized Metropolis acceptance rates which do not apply to SGMCMC).

## 4.2 PARALLEL SGMCMC (AS BAYESIAN DEEP ENSEMBLES)

Deep ensembles (Lakshminarayanan et al., 2017) represent a simple to implement and parallelizable route to uncertainty quantification via multiple runs of gradient descent with different random seeds (for initialization and batch shuffling). Thus, the arguments above point to the formulation of Bayesian deep ensembles, enabled by `posteriors`, that converts the gradient descent ODE equation 4 typically used for deep ensemble training into a Bayesian version that is stationary for the posterior distribution by setting $\mathcal{T} = N^{-1}$, i.e. adding back the noise term in equation 3. Assuming D and Q are chosen to be independent of $z$ this becomes

$$dz = [\mathrm{D} + \mathrm{Q}]N^{-1}\nabla \log \pi(z)dt + \sqrt{2N^{-1}\,\mathrm{D}}dw. \quad (5)$$

The core difference between SGMCMC and the Bayesian deep ensemble or parallel SGMCMC implementation above is that in SGMCMC, samples are collected from a single trajectory, whereas in parallel SGMCMC, multiple trajectories are simulated with only the last sample being retained (although multiple samples per trajectory can also be saved (Margossian et al., 2021)). Parallel SGMCMC provides an asymptotically unbiased approximation to $\pi(z)$ and therefore $p(\theta \mid y_{1:N})$ as long as the criteria discussed in Appendix A.3 are met for a single chain.

---

[1] Equivalently we can rescale $dt \to \mathcal{T}\,dt$ as Brownian motion ($dw$) scales with $\sqrt{t}$, see (1.27) in Pavliotis (2014)

We note that in the large data regime, the additional noise term in equation 5 versus equation 4 becomes very small, and so existing deep ensembles (Lakshminarayanan et al., 2017) can already be seen as approximating the posterior in some sense (with a small bias).

The functional framework of `posteriors` (in combination with `torch.func`) allows easy parallelization of the above unified SGMCMC and SGD algorithms into a parallel SGMCMC implementation that generalizes deep ensembles with minimal code change.

Figure 2 visualizes the behaviour of the techniques discussed in this section for a two-dimensional multi-modal double well posterior. We see clearly that although a deep ensemble finds all modes, it offers no within-mode uncertainty. In contrast, serial SGMCMC explores well within modes but struggles to transition between modes. Parallel SGMCMC combines the benefits of both methods to provide a more representative posterior approximation. In more realistic high dimensional, large $N$ settings runtime and memory requirements become a key consideration. Serial SGMCMC requires less total computation since it runs only a single trajectory. However, the trajectory must be significantly longer and cannot be parallelized; therefore, serial SGMCMC is perhaps not as well suited to modern computational architectures as the deep ensemble or parallel SGMCMC approaches. However, parallel SGMCMC naturally shares many of the caveats of deep ensembles. Notably, the resource requirements are significant to parallelize trajectories, although we note the opportunity to collect multiple samples per trajectory (Margossian et al., 2021). Another point of note is that communication costs between cores are expensive and hyperparameter tuning may require alternative approaches to the serial setting and typically need more care (than optimization) to preserve the correct target distribution across trajectories.

## 5 EXPERIMENTS

We now present experiments investigating the Bayesian learning benefits motivated in Section 1. The functional and swappable nature of `posteriors` makes these experiments particularly accessible across a range of Bayesian learning algorithms. Notably, its scalability and composability with the PyTorch ecosystem allow us to apply Bayesian learning to a host of pre-trained HuggingFace models (Wolf et al., 2020). Reproducible code for all experiments (and more) can be found in the examples folder of the `posteriors` repository.

### 5.1 COLD POSTERIOR EFFECT

This experiment fully utilizes `posteriors` swappability between algorithms to investigate the generalization capabilities of Bayesian learning. As seen in Figure 3, within a `posteriors` implementation the majority of the code is algorithm agnostic and we can easily swap approaches by only changing the build of the `transform` object and its configuration parameters. Extensive experimental details can be found in Appendix D.

The cold posterior effect (Wilson and Izmailov, 2020; Wenzel et al., 2020; Aitchison, 2020; Izmailov et al., 2021) is a phenomenon that has been observed for approximate Bayesian machine learning models where increased predictive performance has been observed when targeting $p_T(\theta \mid y_{1:N}) \propto p(\theta \mid y_{1:N})^{\frac{1}{T}}$ (as discussed in Section 4 with $\mathcal{T} = TN^{-1}$) for $T < 1$, which appears at odds with the Bayesian paradigm (Aitchison, 2020).

Following Wenzel et al. (2020); Izmailov et al. (2021), we train a CNN-LSTM model with 2.7 million parameters on the IMDB (Maas et al., 2011) for binary classification of positive or negative reviews. As in Izmailov et al. (2021), we use a diagonal Gaussian prior with variance $1/40$ for all methods.

The test set loss performance across a range of temperatures for Laplace, VI and SGMCMC approaches are depicted in Figure 4, alongside the baseline optimization MAP approach. See Appendix D for the same plots for accuracy and also optimization for the MLE (which overfits and performs significantly worse than MAP). There are several takeaways from Figure 4, which we describe now.

The Bayesian methods, particularly VI and SGHMC can significantly outperform gradient descent for the MAP when evaluating on the test data. A simple Laplace approximation performs badly, however, improved performance is gained via the linearization discussed in Section 2. Laplace GGN, which is equivalent to $F_C$ in this case (and in most machine learning models, see Appendix A.1),

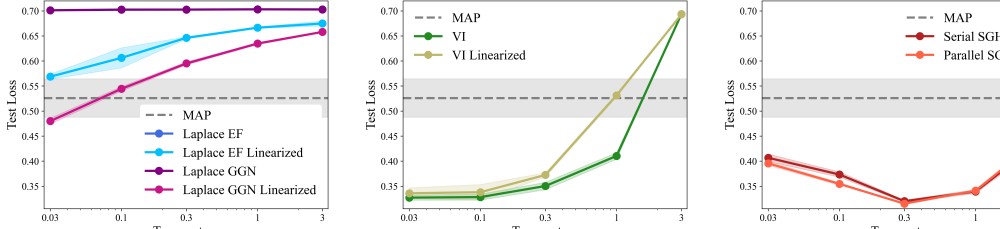

Figure 4: **Investigation into the cold-posterior effect** for a range of `posteriors` algorithms for a CNN-LSTM model (Wenzel et al., 2020) on the IMDB dataset (Maas et al., 2011). Non-linearized Laplace EF and Laplace GGN are indistinguishable (left panel). All Gaussian approximations are diagonal. All approaches display error bars with one standard deviation over 5 random seeds.

improves over $F_E$, which is in agreement with Kunstner et al. (2019). We observe that the reverse is true for VI and that the linearization is actually detrimental. We posit that this is due to the VI optimization acting in parameter space without knowledge of forward linearization. We observe a strong cold posterior effect for the Gaussian approaches with significantly superior performance achieved at $T < 1$. However, for the (non-Gaussian) SGMCMC approaches, only a very mild cold posterior approach is observed. This is evidence in agreement with Izmailov et al. (2021) that the cold posterior effect is more a result of the crudeness of posterior approximations than the of Bayesian learning itself equation 1. Finally, we observe that the parallel SGHMC approach (Bayesian deep ensemble) slightly outperforms the single trajectory serial approach, relatably Figure 2. And further, reducing the temperature deteriorates performance (observing from Section 4 and Appendix C that traditional deep ensemble is recovered as parallel SGHMC with $T = 0$).

## 5.2 CONTINUAL LEARNING WITH LoRA

The purpose of this experiment is to apply a simple Bayesian technique made accessible by `posteriors`, namely, an empirical Fisher Laplace approximation, to mitigate catastrophic forgetting in an online setting with a large-scale model. Here, the composability of `posteriors`, as well as the flexibility of the functional framework, is vital to make use of parameter-efficient fine-tuning (PEFT) (Mangrulkar et al., 2022) and the Llama 2 (Touvron et al., 2023) large language model from the PyTorch (Paszke et al., 2019) ecosystem. Extensive details can be found in Appendix E.

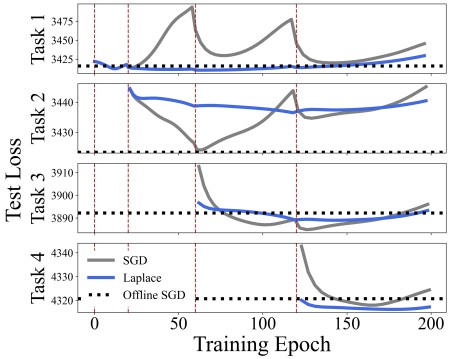

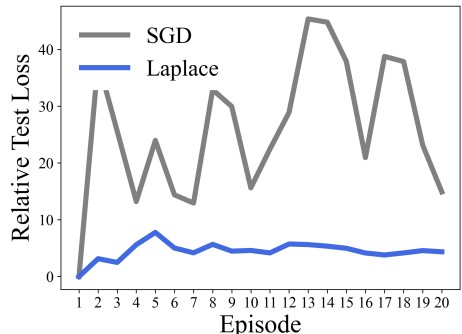

Figure 5: **Continual learning with Llama 2**. The online SGD and Laplace methods train one book after another, whilst the Offline SGD approach sees all books simultaneously, representing the network's capacity. Vertical dashed lines represent episode changes.

Figure 6: **Continual performance averaged over episodes seen thus far.** We use relative loss from a baseline (SGD trained on a single episode) for fairer averaging across episodes.

We consider the problem of episodal continual learning (Kirkpatrick et al., 2017; Nguyen et al., 2017) where each episode's data, $y_n$, is one of 20 books from the PG19 (Rae et al., 2019) dataset with 15%

of each book held out for testing. We compare simple online gradient descent with an approximation to the Bayesian online update equation 1 $p(\theta \mid y_{1:n}) \propto p(\theta \mid y_{1:n-1})p(y_n \mid \theta)$ in Figures 5 and 6. Our baseline implements AdamW via `torchopt`'s integration within the posteriors unified API. The Laplace posterior from the previous episode is then used as the prior for the next episode following the suggested amendment to the EWC method (Kirkpatrick et al., 2017) recommended in (Huszár, 2018). We also train an offline version of the same model with access to all books to indicate the model's total capacity (although unrealistic of an online pipeline).

We use LoRA (Hu et al., 2021) to fine-tune the query, key and output weight matrices in the last attention layer of the 7B parameter model Llama 2 model (Touvron et al., 2023). LoRA computes a low-rank approximation to the weight updates and is implemented in the PEFT library Mangrulkar et al. (2022), which integrates easily into functional `posteriors` code. We use standard choices for the rank ($r = 8$) and scaling ($\alpha = 32$) parameters.

Figure 5 shows test loss for each episode, over all four episodes. The Laplace method maintains low loss in early tasks, while SGD forgets. For example, in the top row, the Laplace approximation stays low throughout training demonstrating that it continues to perform well on task 1 even though it is now being trained on data from tasks 2-4. In contrast, continually applying gradient descent quickly decreases the model's performance on task 1. Figure 6 shows the difference of test loss from baseline, averaged over all episodes seen thus far. We see that the Laplace approximation approach maintains low loss averaged over all tasks.

### 5.3 BAYESIAN LLAMA 3

We now show how `posteriors` can be used to fine-tune a very recent large language model to provide the capability to decompose uncertainty (equation 2) and provide improved out-of-distribution detection. The flexible and scalable design of `posteriors` allows us to easily use subspace methods for fine-tuning a large pre-trained model in parallel using the novel Bayesian generalization of deep ensembles introduced in Section 4.2. Additionally, `posteriors`' functional approach allows us to easily and efficiently map over the ensemble in forward calls at inference time. Extensive experimental details can be found in Appendix F.

We fine-tune the last attention layer of the 8B Llama 3 model (AI@Meta, 2024) (resulting in 218 million trainable parameters) on the TQA (Kembhavi et al., 2017) dataset, which consists of scientific textbooks. We fine-tune a baseline with SGD and a size 10 ensemble trained in parallel with `posteriors`' SGHMC. We also compare against the base Llama 3 model without fine-tuning. We evaluate the models by asking them to complete 100 scientific statements (in-distribution) with the last word removed. We then repeat the experiment with the statements translated into Samoan, an out-of-distribution language for TQA (Kembhavi et al., 2017).

We can see from Figure 7 that although total uncertainty provides some indication for classifying the Samoan text as out-of-distribution, the epistemic uncertainty of parallel SGMHC offers a clearer split. Table 1 validates this with the AUROC metric (Kuhn et al., 2023), whilst also displaying the categorical loss values for the next token of the English statements to verify a successful fine-tune.

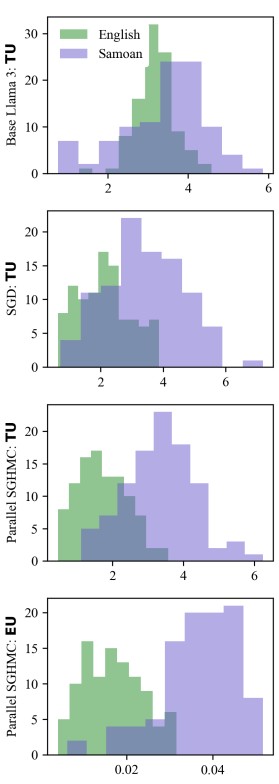

Figure 7: **Distributions of predictive uncertainties** with Llama 3.

Overall, we can see a significant advantage in out-of-distribution detection via the Bayesian decomposition of uncertainty, whereas the point-estimate-based methods (MLE/MAP) cannot decompose uncertainty to differentiate epistemic uncertainty from aleatoric or semantic uncertainty which is key for open-ended natural language generation (Kuhn et al., 2023).

In Appendix F, we include results for serial SGHMC, which interestingly fails to provide improved epistemic uncertainty quantification. We posit that this has to do with serial SGHMC staying isolated in a single mode of the posterior in this case, and therefore, averaging along the trajectory provides little over an SGD approach as in Figure 2.

Table 1: **Out-of-distribution detection** with Bayesian Llama 3.

|  | Base Llama 3 TU | SGD TU | Parallel SGHMC TU | EU |
|---|---|---|---|---|
| **AUROC ↑** | 0.62 | 0.80 | 0.90 | **0.95** |
| **Loss on English statements ↓** | 4.68 | **4.20** | 4.40 | |

## 6 RELATED WORK

Section 2 (and more thoroughly Appendix A) summarizes much of the related work regarding Bayesian methodology. Additional comprehensive reviews can be found in Wang and Yeung (2020) with Fortuin (2022) focussing on prior specifications (which we do not go into detail here but is handled with broad flexibility in `posteriors`) and Izmailov et al. (2021) providing extensive experiments with full-batch MCMC to give a closer indication of the behaviour of the true Bayesian posterior.

There is a broad selection of software for uncertainty quantification; however, to our knowledge and experience, none satisfy all five points in Section 3. Stan (Carpenter et al., 2017) is widely used within the field of Bayesian statistics however does not support minibatching and is not easily extensible. Pyro (Bingham et al., 2019) is a broad framework that supports a range of models and inference techniques; however, it is more heavyweight than `posteriors` and harder to compose with existing PyTorch code. BlackJAX (Cabezas et al., 2024) (and optax (DeepMind et al., 2020)) heavily inspired the `posteriors` functional approach. However, BlackJAX also has a primary focus on full-batch inference whilst `posteriors` is minibatch-first and, like other JAX (Phan et al., 2019; Detommaso et al., 2023; Duffield, 2021) packages, does not compose with the PyTorch ecosystem.

## 7 CONCLUSION AND OUTLOOK

We have presented a new software package `posteriors` for scalable and extensible uncertainty quantification with PyTorch (Paszke et al., 2019) models. Through numerical experiments, we have demonstrated how `posteriors` can be used to achieve a range of benefits over optimization-based approaches and composing with PyTorch toolkits. In particular, we highlighted that the cold posterior effect mostly occurs in Gaussian approximations to the Bayesian posterior as well as demonstrating the benefits of Bayesian learning in the very practical and large-scale setting of free-form text generation with large language models.

We are excited by the future prospects for `posteriors`. Its functional API makes it easy to maintain and extend; we already have a long list of methods and approaches to add. `posteriors` is fully open source, and we are keen to foster a community for feedback for feature requests and encourage contributions. The framework from Section 4 allows for simultaneous research into optimization and sampling methods. Combining that with `posteriors`' broad range of utility functions (that, to our knowledge, are not available elsewhere in the PyTorch ecosystem), means that implementation and research into second-order techniques (Martens et al., 2010; Martens, 2020) and improved discretization methods (Leimkuhler et al., 2018) represent accessible avenues for future work in both optimization and SGMCMC.

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

## A  SURVEY OF SCALABLE BAYESIAN LEARNING

We now provide a more thorough review of methods for scalable Bayesian learning, extending Section 2.

### A.1  LAPLACE APPROXIMATION

As discussed, the Laplace approximation (MacKay, 1992; Daxberger et al., 2021) forms a Gaussian approximation to the posterior distribution $p(\theta \mid y_{1:N}) \approx \mathbf{N}(\theta \mid \theta_{\text{MAP}}, \hat{\Sigma})$ based on the second order Taylor expansion around the posterior mode $\theta_{\text{MAP}} = \arg\max_\theta p(\theta \mid y_{1:N})$:

$$\log p(\theta \mid y_{1:N}) \approx (\theta - \theta_{\text{MAP}})^\top \nabla_\theta^2 \log p(\theta_{\text{MAP}} \mid y_{1:N})(\theta - \theta_{\text{MAP}}) + \text{const},$$

$$\approx \log \mathbf{N}(\theta \mid \theta_{\text{MAP}}, \hat{\Sigma}) + \text{const}.$$

where $\nabla_\theta \log p(\theta \mid y_{1:N}) = 0$ at $\theta = \theta_{\text{MAP}}$.

The Taylor expansion implies the use of the negative inverse Hessian $-\nabla_\theta^2 \log p(\theta_{\text{MAP}} \mid y_{1:N})^{-1}$ as the covariance $\hat{\Sigma}$, however this is not guaranteed to be positive semidefinite and is therefore invalid

as a covariance matrix. Instead, we can draw from the Bernstein-von Mises theorem (Hartigan and Hartigan, 1983):

$$p(\theta \mid y_{1:N}) \underset{N \to \infty}{\to} \mathbf{N}\left(\theta \mid \theta_{\text{MAP}}, N^{-1} \, \mathrm{F}(\theta_{\text{MAP}})^{-1}\right),$$

with Fisher information matrix (which is guaranteed to be positive semidefinite):

$$\mathrm{F}(\theta) = \mathbb{E}_{p(y\mid\theta)}[\nabla_\theta^2 \log p(\theta \mid y)] = \mathbb{E}_{p(y\mid\theta)}[\nabla_\theta \log p(\theta \mid y)\nabla_\theta \log p(\theta \mid y)^\top]. \tag{6}$$

In the supervised learning setting with inputs $x$, the Fisher information matrix becomes $\mathrm{F}(\theta) = \mathbb{E}_{p(x,y\mid\theta)}[\nabla_\theta^2 \log p(\theta \mid x, y)]$ which is intractable as we do not have access to the distribution over inputs $p(x)$. Two popular approximations use the training data $\hat{p}(x,y) = N^{-1}\sum_{i=1}^N \delta(x,y \mid x^i, y^i)$. The *empirical Fisher*:

$$\mathrm{F}_{\mathrm{E}}(\theta) = \mathbb{E}_{\hat{p}(x,y)}[\nabla_\theta \log p(\theta \mid y)\nabla_\theta \log p(\theta \mid y)^\top],$$

and *conditional Fisher*:

$$\mathrm{F}_{\mathrm{C}}(\theta) = \mathbb{E}_{\hat{p}(x)p(y\mid x,\theta)}[\nabla_\theta \log p(\theta \mid y)\nabla_\theta \log p(\theta \mid y)^\top], \tag{7}$$

with the conditional Fisher generally recommended (Kunstner et al., 2019).

The integral with respect to $p(y \mid x, \theta)$ in equation 7 may initially seem difficult to compute. Fortunately, things are simplified in the common machine learning setting where the likelihood has the form $p(y \mid x, \theta) = p(y \mid f_\theta(x))$ where $p(y \mid z)$ is the probability density function for an exponential family distribution with natural parameter $z$. That is

$$\log p(y \mid z) = z^\top T(y) - \log \int z^\top T(y) dy,$$

for some $T(y)$. This setting includes the common cases of classification (softmax with cross-entropy loss) and regression (mean squared error loss). Then we have an exact equivalence between $F_C(\theta)$ and the Generalized Gauss-Newton matrix (GGN, Martens (2020)) matrix:

$$G(\theta) = \mathbb{E}_{\hat{p}(x,y)}[\nabla_\theta f_\theta(x)^\top H_L(x,y)\nabla_\theta f_\theta(x)],$$

where $\nabla_\theta f_\theta(x)$ is the Jacobian of the forward function and

$$H_L(x,y) = [-\nabla_z^2 p(y \mid z)]_{z=f_\theta(x)},$$

is the Hessian of the loss function evaluated at $z = f_\theta(x)$.

### CAVEATS AND COMPUTATIONAL BOTTLENECKS

Laplace approximation training consists of first determining the posterior mode $\theta_{\text{MAP}}$ through gradient descent. This is followed by a single additional epoch to calculate the approximate covariance. It is, thus, cheap to compute in the large data regime. However, in the large parameter regime, storing the covariance matrix with $\mathcal{O}(d^2)$ elements becomes a memory bottleneck as well as the required inverting and sampling operations, which typically have runtime $\mathcal{O}(d^3)$. As such, approximations are used, such as diagonal (MacKay, 1992), K-FAC (Ritter et al., 2018) and low-rank (Daxberger et al., 2021) with reduced $\mathcal{O}(d)$ memory. The main caveat for Laplace approximations is the accuracy of a Gaussian approximation under the extreme non-linearities in a neural network posterior (Izmailov et al., 2021), which applies even further to reduced memory approximations.

### A.2 VARIATIONAL INFERENCE

Variational inference seeks a parameterized distribution $q_\phi(\theta)$ that minimizes some distance from the target posterior distribution $p(\theta \mid y_{1:N})$. Most commonly (Blei et al., 2017) we assume a Gaussian variational distribution $q_\phi(\theta) = \mathbf{N}(\theta \mid \mu, \Sigma)$ with $\phi = (\mu, \Sigma)$ and the distance to be minimized is the KL divergence:

$$0 \le \mathrm{KL}[q_\phi(\theta) \mid\mid p(\theta \mid y_{1:N})] = \log p(y_{1:N}) + \underbrace{\mathbb{E}_{q_\phi(\theta)}\left[\log \frac{q_\phi(\theta)}{p(\theta)} - \log p(y_{1:N} \mid \theta)\right]}_{\text{NELBO}(\phi)}. \tag{8}$$

Thus, the problem becomes one of optimization to minimize the *negative evidence lower bound* objective $\mathrm{NELBO}(\phi)$, which can be achieved with stochastic gradient descent alongside the reparameterization trick (Rezende et al., 2014) to obtain unbiased gradients of the expectation.

CAVEATS AND COMPUTATIONAL BOTTLENECKS

Notably, the NELBO$(\phi)$ objective can be estimated stochastically by sampling from $q_\phi(\theta)$ and even minibatching the data $y_{1:N}$. Therefore, training is scalable in the large data regime. The caveat regarding the crudeness of a Gaussian approximation to the posterior also applies here, except that variational inference minimizes a formal divergence from the posterior rather than relying on Bernstein-von Mises asymptotics. However, in the large parameter regime, the NELBO$(\phi)$ objective requires high dimensional integration at every optimization step, which can suffer from high variance even with the available tricks (Blei et al., 2017).

## A.3 Stochastic gradient Markov chain Monte Carlo

A completely different paradigm to posterior approximation is to construct a Monte Carlo approximation $p(\theta \mid y_{1:N}) \approx N^{-1} \sum_{k=1}^{K} \delta(\theta \mid \theta_k)$. This is most commonly achieved by evolving a stochastic differential equation (SDE) and collecting samples along the trajectory. Following Ma et al. (2015), an SDE has $\pi(z)$ as a stationary distribution (and therefore provides an asymptotically unbiased Monte Carlo approximation) if and only if it is of the form

$$dz = [\mathrm{D}(z) + \mathrm{Q}(z)]\nabla \log \pi(z)dt + \nabla \cdot [\mathrm{D}(z) + \mathrm{Q}(z)]dt + \sqrt{2\,\mathrm{D}(z)}dw. \tag{9}$$

with symmetric positive-semidefinite $\mathrm{D}(z)$ and skew symmetric $\mathrm{Q}(z) = -\mathrm{Q}(z)^\top$. Here $z = (\theta, \omega)$ where $\omega$ are any auxiliary parameters such as momenta and thermostat (Ma et al., 2015; Ding et al., 2014). The distribution $\pi(z)$ is thus an extended target with $p(\theta \mid y_{1:N})$ marginal in $\theta$. The matrices $\mathrm{D}(z)$ and $\mathrm{Q}(z)$ are often chosen as independent of $z$, meaning the matrix divergence term vanishes.

The exact simulation of such a non-linear SDE is intractable, and as such, approximate discretization schemes such as the Euler-Maruyama (Welling and Teh, 2011) method are employed. Traditional full-batch Markov chain Monte Carlo (MCMC) schemes (Andrieu et al., 2003) use a Metropolis-Hastings step to ensure convergence of the Monte Carlo approximation is preserved through discretization. In the minibatch setting, the Metropolis-Hastings ratio cannot be estimated easily. Instead, we can use an unbiased stochastic gradient $\widehat{\nabla \log \pi}(z) \sim \mathrm{N}(\cdot \mid \nabla \log \pi(z), \mathrm{B}(z))$, then as noted in Welling and Teh (2011) the stochasticity arising from minibatching, $\mathrm{B}(z)$, decreases like $\mathcal{O}(\epsilon_t^2)$ for learning rate $\epsilon_t$. A Robbins-Monro (Robbins and Monro, 1951) learning rate schedule $\sum \epsilon_t^2 < \sum \epsilon_t = \infty$ ensures an asymptotically unbiased Monte Carlo approximation (Welling and Teh, 2011). Additional enhancements can be made if an approximation to $\mathrm{B}(z)$ is available (Ma et al., 2015).

We detail specific implementations of SGMCMC in Section B.

CAVEATS AND COMPUTATIONAL BOTTLENECKS

The stochastic gradient formulation of MCMC permits easy application to the minibatch and large data regime. However, the decreasing learning rate requirement results in diffusive behaviour (Alexos et al., 2022; Li et al., 2016; Bieringer et al., 2023) where the discretized SDE puts excessive weight on the noise term $dw$ over the informative gradient term. This is problematic since exploration is very much desired for MCMC. Often, a constant learning rate is used in practice (Leimkuhler et al., 2018) with an asymptotic bias incurred. In the large parameter regime, the formulation of a Monte Carlo approximation becomes a memory bottleneck. Typical lower dimensional MCMC procedures take many thousands of samples, but due to memory constraints, only a significantly smaller sample size is available for large neural network parameters, resulting in a crude approximation. A notable further caveat is that, in contrast to the Gaussian approximations, the resulting Monte Carlo approximation does not provide access to a density with pointwise evaluations and cannot be used as a prior for subsequent Bayesian updates in an online regime equation 1.

## A.4 Notable mentions

Perhaps the most common route to uncertainty quantification in neural networks is dropout (Gal and Ghahramani, 2016; Gal et al., 2017), where model parameters or neurons are randomly removed from each forward call. This approach can be effective and has a Bayesian interpretation in the posterior probability space of inclusion/exclusion, but this is naturally more restricted than the posterior approximations above.

Stochastic weight averaging (SWA, Izmailov et al. (2018)) and SWA-Gaussian (SWAG, Maddox et al. (2019)) are similar in spirit to SGMCMC as they use samples from a single trajectory to generate a point or Gaussian approximation to the posterior respectively. Another option for formulating a Gaussian distribution over parameters is to consider each minibatch as a small-scale Bayesian update; local linearization of this update is the paradigm of the extended Kalman filter (Duran-Martin et al., 2022), which also has natural gradient descent (Ollivier, 2018) and variational inference interpretations (Zhang et al., 2018). These methods are easy to implement but again suffer from the caveats of Gaussian posterior approximations above.

## B    SGMCMC ALGORITHMS

Recall the generic SDE Ma et al. (2015) from Section 4 for targeting the tempered $\pi_{\mathcal{T}}(z) \propto \pi(z)^{\frac{1}{N\mathcal{T}}}$ where we set $\mathcal{T} = N^{-1}$ to target the Bayesian posterior

$$dz = [\mathrm{D}(z) + \mathrm{Q}(z)]N^{-1}\nabla \log \pi(z)dt + \mathcal{T}\nabla \cdot [\mathrm{D}(z) + \mathrm{Q}(z)]dt + \sqrt{2\,\mathcal{T}\,\mathrm{D}(z)}dw,$$

for symmetric positive-semidefinite $\mathrm{D}(z)$ and skew symmetric $\mathrm{Q}(z) = -\mathrm{Q}(z)^\top$. With $z = (\theta, \omega)$ for any auxiliary variables $\omega$ and $\int \pi(\theta, \omega)d\omega = \pi(\theta) = p(\theta \mid y_{1:N})$.

The matrix divergence $\nabla \cdot \mathrm{M}(z) \in \mathbf{R}^d$ is defined Ma et al. (2015) as

$$[\nabla \cdot \mathrm{M}(z)]_i = \sum_{j=1}^{d} \frac{\partial}{\partial z_j}[\mathrm{M}(z)]_{ij},$$

and vanishes in the common case that the preconditioners are set to be independent of $z$, i.e. $\mathrm{D}(z) = D$ and $\mathrm{Q}(z) = Q$.

We now cover the most popular SGMCMC algorithms and those implemented in `posteriors`, all of which are also found in Ma et al. (2015).

### B.1    SGLD

The simplest SGMCMC algorithm is stochastic gradient Langevin dynamics (Welling and Teh, 2011), which contains no auxiliary variables (i.e. $z = \theta$) and is represented by $\mathrm{D} = \mathbb{I}$ and $\mathrm{Q} = 0$. This gives continuous-time SGLD:

$$d\theta = N^{-1}\nabla \log \pi(\theta)dt + \sqrt{2\mathcal{T}}dw.$$

And Euler-Maruyama discretization:

$$\theta_{k+1} = \theta_k + \epsilon N^{-1}\nabla \log \pi(\theta_k) + \sqrt{2\epsilon\mathcal{T}}\zeta_k, \qquad\qquad \zeta_k \sim \mathbf{N}(\zeta \mid 0, \mathbb{I}).$$

### B.2    SGHMC

Stochastic gradient Hamiltonian Monte Carlo Chen et al. (2014) extends $z = (\theta, m)$ to include momenta $m$ with Gaussian target distribution

$$\log \pi(\theta, m) = \log \pi(\theta) - \frac{1}{2\sigma^2}m^\top m + \mathrm{const.}$$

Choosing $\mathrm{D} = \begin{pmatrix} 0 & 0 \\ 0 & \alpha\mathbb{I} \end{pmatrix}$ and $\mathrm{Q} = \begin{pmatrix} 0 & -\mathbb{I} \\ \mathbb{I} & 0 \end{pmatrix}$ gives continuous-time SGHMC

$$d\theta = \sigma^{-2}mdt,$$
$$dm = N^{-1}\nabla \log \pi(\theta)dt - \alpha\sigma^{-2}mdt + \sqrt{2\,\mathcal{T}\,\alpha}dw,$$

and Euler-Maruyama discretization:

$$\theta_{k+1} = \theta_k + \epsilon\sigma^{-2}m_k,$$
$$m_{k+1} = m_k + \epsilon N^{-1}\nabla \log \pi(\theta) - \epsilon\sigma^{-2}\alpha m_k + \sqrt{2\epsilon\mathcal{T}\alpha}\zeta_k, \qquad \zeta_k \sim \mathbf{N}(\zeta \mid 0, \mathbb{I}).$$

### B.3 SGNHT

The stochastic gradient Nosé-Hoover thermostat (SGNHT) algorithm (Ding et al., 2014) uses ideas from thermodynamics (Leimkuhler et al., 2018) to introduce an additional scalar variable $\xi$ that automates the selection of the $\alpha$ friction parameter in SGHMC. We have $z = (\theta, m, \xi)$ targeting [2]

$$\log \pi(\theta, m, \xi) = \log \pi(\theta) - \frac{1}{2\sigma^2} m^\top m - \frac{d}{2}(\xi - \alpha)^2 + \text{const.}$$

Choosing

$$\text{D}(\theta, m, \xi) = \begin{pmatrix} 0 & 0 & 0 \\ 0 & \alpha\mathbb{I} & 0 \\ 0 & 0 & 0 \end{pmatrix}, \quad \text{Q}(\theta, m, \xi) = \begin{pmatrix} 0 & -\mathbb{I} & 0 \\ \mathbb{I} & 0 & m/d \\ 0 & -m^\top/d & 0 \end{pmatrix},$$

gives continuous-time SGNHT

$$d\theta = \sigma^{-2} m dt,$$
$$dm = N^{-1}\nabla \log \pi(\theta)dt - \xi\sigma^{-2}mdt + \sqrt{2\mathcal{T}\alpha}dw,$$
$$d\xi = (\sigma^{-2}d^{-1}m^\top m - \mathcal{T})dt,$$

and Euler-Maruyama discretization:

$$\theta_{k+1} = \theta_k + \epsilon\sigma^{-1}m_k,$$
$$m_{k+1} = m_k + \epsilon N^{-1}\nabla \log \pi(\theta) - \epsilon\sigma^{-2}\xi_k m_k + \sqrt{2\epsilon\mathcal{T}\alpha}\zeta_k, \qquad \zeta_k \sim \mathbf{N}(\zeta \mid 0, \mathbb{I}),$$
$$\xi_{k+1} = \xi_k + \epsilon\left(\sigma^{-2}d^{-1}m_k^\top m_k - \mathcal{T}\right).$$

Although SGNHT has the same tuning parameters $(\epsilon, \sigma, \alpha)$ as SGHMC, we note the $\alpha$ parameter only controls the noise level for the $m$ update and is absent from the deterministic dynamics, with even $\alpha = 0$ representing a reasonable choice.

## C SGMCMC FOR OPTIMIZATION

Let us consider gradient descent (with momenta) (Sutskever et al., 2013), parameterization from PyTorch, for minimising $U(\theta)$:

$$\theta_{k+1} = \theta_k + \gamma m_k,$$
$$m_{k+1} = \mu m_k - (1 - \tau)\nabla U(\theta_k).$$

Here vanilla SGD $\theta_{k+1} = \theta_k - \gamma\nabla U(\theta_k)$ is obtained with $\mu = 0$ and $\tau = 0$.

And also SGHMC B.2 with $\mathcal{T} = 0$ for $U(\theta) = -N^{-1}\log \pi(\theta) = -N^{-1}\log p(\theta \mid y_{1:N})$:

$$\theta_{k+1} = \theta_k + \epsilon\sigma^{-2}m_k,$$
$$m_{k+1} = (1 - \epsilon\sigma^{-2}\alpha)m_k - \epsilon\nabla U(\theta_k).$$

We see an equivalence between SGD and SGHMC ($T = 0$) (Wenzel et al., 2020) with

$$
\begin{array}{lcl lcl}
\gamma & = & \epsilon\sigma^{-2}, & \epsilon & = & 1 - \tau, \\
\mu & = & 1 - \epsilon\sigma^{-2}\alpha, \quad\Longleftrightarrow\quad & \sigma^{-2} & = & \gamma(1 - \tau)^{-1}, \\
\tau & = & 1 - \epsilon, & \alpha & = & (1 - \mu)\gamma^{-1}.
\end{array} \tag{10}
$$

We note that the transition from SGMCMC to optimization and back is not ubiquitous amongst methods. Indeed, the Euler-Maruyama SGNHT sampling algorithm B.3, does not result in a functional optimization algorithm at the zero temperature limit as the value thermostat can only increase and results in divergant dynamics. Also, the popular Adam (Kingma and Ba, 2014) optimizer does not fit the general sampling SDE equation 9 due to its use of elementwise operations, which do not have a convenient matrix-vector-product form. `posteriors`' framework makes these mathematical considerations transparent and an easily accessible avenue for research.

---

[2]Note that the SGNHT target distribution for $\xi$ is erroneously claimed as $\frac{1}{2d}(\xi - \alpha)^2$ in Ma et al. (2015) and Nemeth and Fearnhead (2021)

## D  EXPERIMENT DETAILS: COLD POSTERIOR EFFECT

We now give additional details for the experiment into generalization capabilities of the Bayesian methods in `posteriors` and the cold posterior effect, from Section 5.1. Fully reproducible code can be found at `posteriors/examples/imdb`.

We here display more details on the results displayed in Figure 4. Figure 8 is exactly the same as Figure 4, but with the test loss for the MLE, approach added (no prior), which severely overfits. Figure 9 reports all accuracies which agree with the takeaways discussed in Section 5.1.

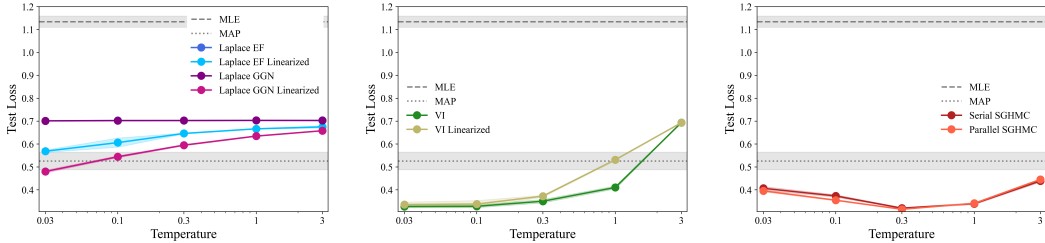

Figure 8: **Cold posterior test losses** (lower is better) for a range of `posteriors` algorithms and temperatures for the CNN-LSTM model (Wenzel et al., 2020) on the IMDB dataset (Maas et al., 2011) from Section 5.1 (same as Figure 4 with MLE added). Laplace EF is indistinguishable from Laplace GGN.

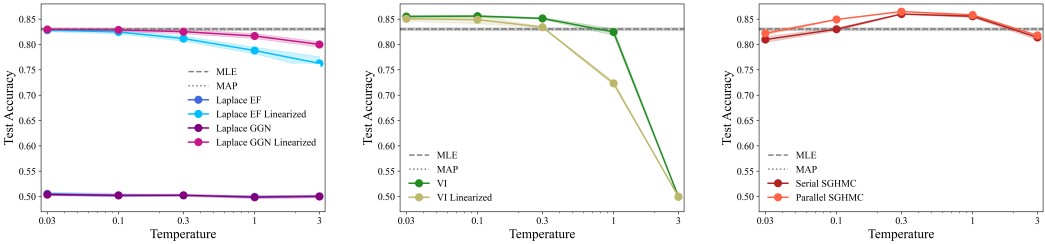

Figure 9: **Cold posterior test accuracies** (higher is better) for a range of `posteriors` algorithms and temperatures for the CNN-LSTM model (Wenzel et al., 2020) on the IMDB dataset (Maas et al., 2011) from Section 5.1.

We use exactly the same model CNN-LSTM from Wenzel et al. (2020); Izmailov et al. (2021), which consists of an embedding layer to convert input sequences into dense vectors, a 1D convolutional layer to capture local features, ReLU activation and max-pooling layers, an LSTM layer to capture long-term dependencies, and a fully connected layer to produce the final classification logits. The model has 2.7 million trainable parameters. Following Izmailov et al. (2021) we use a diagonal Gaussian prior with all variances set to 1/40 (aside from MLE which has no prior or equivalently infinite prior variances).

We train on the IMDB Maas et al. (2011) dataset for binary classification of positive/negative sentiment. We follow the default 50-50 split for the dataset with 25 thousand samples for training and 25 thousand samples for testing. All methods used batch size 32.

All approaches were averaged over 5 random seeds where for parallel SGHMC 5 ensembles were bootstrapped from 35 chains each with their own random seed.

We now detail the hyperparameters used for each method (except for serial SGHMC, all methods were trained for 30 epochs):

- **MLE**: We train using the AdamW optimizer (Loshchilov and Hutter, 2018) with all hyperparameters set to the defaults from TorchOpt (Ren et al., 2023) (learning rate $10^{-3}$).
- **MAP**: Same exact settings as MLE but with finite prior variances (set to 1/40 as with the other methods).

- **Laplace**: We take the trained MAP parameters as the mean of the Gaussian distribution. The diagonal covariances are calculated with `posteriors`, and we compare both empirical Fisher and GGN (which is equivalent to the conditional Fisher for our model). For test set evaluation, we sample 50 samples in parameter space and $10^4$ samples in logit space for the linearized approach.

- **Variational Inference**: As above, we train using the AdamW optimizer (Loshchilov and Hutter, 2018) with all default parameters. We apply the *stick-the-landing* variance reduction (Roeder et al., 2017) and parameterization (Rezende et al., 2014) tricks. We use a single sample Monte Carlo estimate at each step. We train the variational variances in log space to avoid negative variances and initialise all log standard deviations to $-3$. For test set evaluation, we sample 50 samples in parameter space and $10^4$ samples in logit space for the linearized approach.

- **Serial SGHMC**: We train with learning rate $0.1$ and friction $\alpha = 1$. We run for 60 epochs (which is longer as many samples are collected along a single trajectory). We apply a burn-in of 20000 thousand iterations and then collect samples every 1000 iterations, resulting in a final collection of 27 samples.

- **Parallel SGHMC**: Setup was exactly the same as serial except only 30 epochs were trained, and only the final parameters were taken. This was repeated across 35 different random seeds (for data shuffling and SGHMC noise) in parallel, for testing we bootstrap sampled 5 ensembles of size 15 from the 35 chains.

All cold posterior simulations were run on an NVIDIA A100, and all simulations (including repeats over 5 random seeds) take $\sim 1$ day to run.

# E   EXPERIMENT DETAILS: CONTINUAL LEARNING WITH LORA

In this section, we detail details for the continual learning experiment from Section 5.2. Fully reproducible code can be found at `posteriors/examples/continual_lora`.

Our dataset Rae et al. (2019) for the continual learning experiment, a collection of long books, is divided into $N$ episodes of train and test data. In the results reported in Figures 5 and 6, we use 1 book per episode, holding out the last $15\%$. We select the first twenty books from Rae et al. (2019) that satisfy the following criterion:

1. Published in 1900 or later
2. Is at least 100,000 tokens, and at most 1,000,000 tokens

We use LoRA (Hu et al., 2021) on Llama 2 (Touvron et al., 2023) to reduce the dimension of the inference task to 200k trainable parameters. We test two baseline methods. We fine-tune the model on each episode's new data, a standard approach known to result in catastrophic forgetting, and we also implement a static offline baseline that sees all data every episode. This represents the LoRA network's total capacity but is computationally infeasible for online learning.

The prior for the next episode is the posterior from the previous episode, becoming a quadratic penalty in the loss function during gradient descent. Whereas the original work on catastrophic forgetting (Kirkpatrick et al., 2017) suggested using multiple penalties (incorporating data from all previous episodes into the prior), we use a single penalty following Huszár (2018). We use only the last episode's posterior, reminiscent of exact Bayesian updates.

For each episode of data, we perform fine-tuning on the current episode's data and test on for all data episodes seen thus far. The Laplace method additionally updates the posterior based on new MAP estimates of weights, and the latest Fisher information. The posterior is used as the prior for the next episode. These steps are handled by `posteriors`.

Additional Experiment Settings:

1. We use rank $r = 8$ and $\alpha = 32$ for LoRA, the standard settings (Hu et al., 2021), and fine-tune three weight matrices in the last layer (key, query, and output projections), following the literature (Yang et al., 2024).

Table 2: **Out-of-distribution metrics** with Bayesian Llama 3.

|  | Base Llama 3 | SGD | Serial SGHMC | | Parallel SGHMC | |
|---|---|---|---|---|---|---|
|  | **TU** | **TU** | **TU** | **EU** | **TU** | **EU** |
| **AUROC ↑** | 0.62 | $0.80 \pm 0.02$ | 0.85 | 0.17 | 0.90 | **0.95** |
| **Loss on English statements ↓** | 4.68 | $\mathbf{4.20} \pm 0.14$ | 4.28 | | 4.40 | |
| **Loss on Samoan statements ↓** | 11.41 | $9.87 \pm 0.05$ | 9.98 | | **9.83** | |

2. We stride over the texts so that all tokens have 2048 context tokens, given that there are 2048 previous tokens to condition on.

Continual LoRA simulations were run on an NVIDIA A100; simulations over the 20 books take $\sim 6$ hours to run.

## F    EXPERIMENT DETAILS: BAYESIAN LLAMA 3

Here we describe further details for our final experiment fine-tuning the 8B Llama 3 model (AI@Meta, 2024). Fully reproducible code can be found at `posteriors/examples/bayes_llama3`.

As previously mentioned, our experiments involve using 10 sets of ensemble models for the last attention layer of the Llama3 model, resulting in 218 million trainable parameters. We train via SGHMC sampling for 20 epochs over the TQA dataset (Kembhavi et al., 2017). For evaluation, we generated a random set of 100 scientific factual statements from ChatGPT, which we also validated by hand. Additional hyperparameter and training details are as follows:

1. We set the learning rate to be $10^{-3}$. We set alpha and beta parameters to $10^{-2}$ and 0, respectively, with the momenta initialized to 0.

2. For the textbook content, we tokenize with a stride length of 300 and a stride overlap of size 100 while using a batch size of 10. During training, we mask the loss on the first 250 tokens and only consider the loss on the last 50 tokens.

3. All layers are frozen except for the final attention layer.

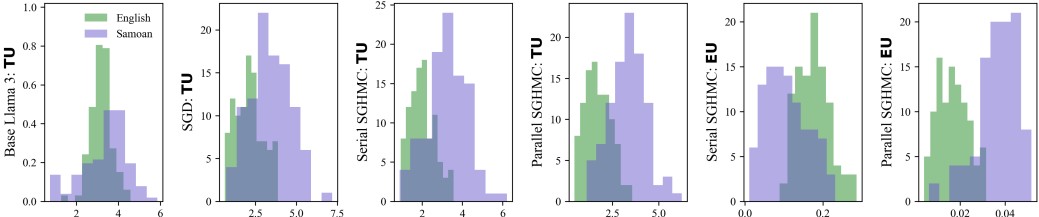

Figure 10: **Distributions of predictive uncertainties** with Llama 3 (Serial SGHMC included).

In Table 2, we extend Table 1 to include the loss results for predicting the last word of the statements translated into Samoan. Where we can clearly see the Samoan text is significantly out of distribution. We also include one standard deviation error from 4 random seeds for SGD.

In Figure 10, we extend Figure 7 to include results for serial SGHMC. We see that serial SGHMC fails to provide a useful measure of epistemic uncertainty, which, as discussed in Section 5.3, we posit is due to serial SGHMC struggling to escape local modes and, therefore, lacking diversity in predictions - and even further, as the collected parameters are very similar (in such high dimension), this gives a misguided indication of low epistemic uncertainty (a large agreement in produced logits) that results in worse performance than using TU.

Bayesian Llama 3 simulations were run on an NVIDIA A100; training of the SGD and ensemble approaches take $\sim 16$ hours to run whilst evaluation over the 100 statements is fast.

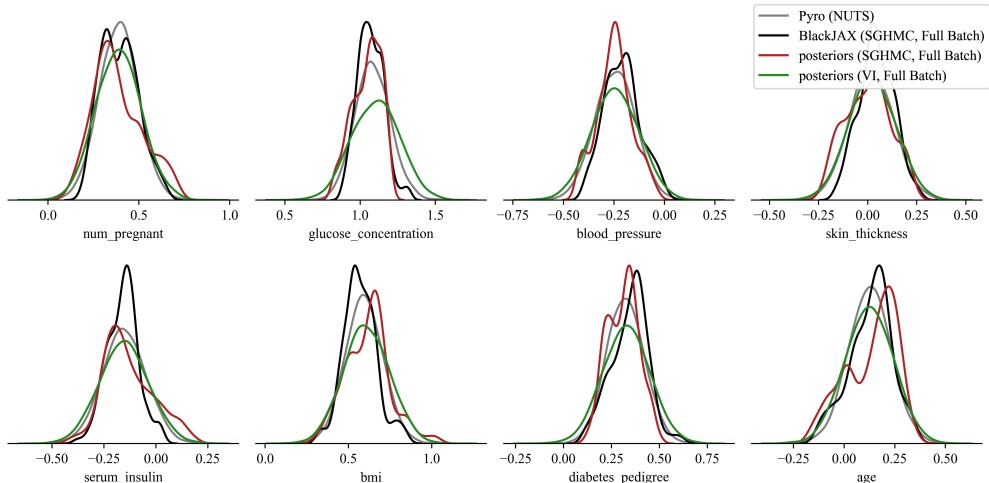

Figure 11: **Full batch marginal distributions for Bayesian logistic regression** on the Pima Indians dataset. All methods generated 5000 samples and provided approximations that are broadly in agreement.

## G   BAYESIAN LOGISTIC REGRESSION: COMPARISON AND COMPOSITION WITH PYRO

In this section, we use the probabilistic programming language (PPL) Pyro (Bingham et al., 2019) to construct a more classical Bayesian logistic regression model on the Pima Indians dataset (Smith et al., 1988) which is a standard benchmark in computational statistics, see e.g. Titsias (2024). The model is constructed in Pyro and seamlessly converted into a `log_posterior` function for scalable inference with `posteriors`. These simulations demonstrate appropriate convergence of the `posteriors` in a smaller-scale setting, compare efficiency to other frameworks and demonstrate seamless composition with a PPL in Pyro.

We compare `posteriors`' SGHMC and VI implementations with Pyro's automated NUTS (Hoffman et al., 2014) implementation (which represents a gold standard) and a JAX (Bradbury et al., 2018) implementation with BlackJAX (Cabezas et al., 2024). All serial MCMC methods were run for 5000 samples with similar warmup and thinning configurations (noting that NUTS has an adaptive thinning procedure). VI was ran for 8000 iterations which was sufficient for convergence. For parallel SGHMC we run 5000 samples independently for 2000 iterations.

We observe, in Figure 11, that all methods (including mini-batched versions, Figure 12) are broadly in agreement and have roughly converged to the same posterior distribution. In Table 3, we compare the fidelity of the posterior approximations by visualising the marginal distributions and comparing quantitatively via a kernelized Stein discrepancy (Gorham and Mackey, 2015). We also display the runtimes of the approaches although in this case the model is small (8 dimensions and 768 data points) and therefore the Python overheads are relatively significant over model calls (as demonstrated by only a marginal speedup from mini-batching) and this is more so for PyTorch over JAX (although this rapidly deteriorates in the case of larger, more expensive model calls, in our experience). Nevertheless, we see that posteriors is competitive with Pyro (Bingham et al., 2019). Importantly, these simulations demonstrate that `posteriors` can easily be composed one of the most popular probabilistic programming languages in Pyro (Bingham et al., 2019) to perform efficient and scalable mini-batch inference in traditional Bayesian models.

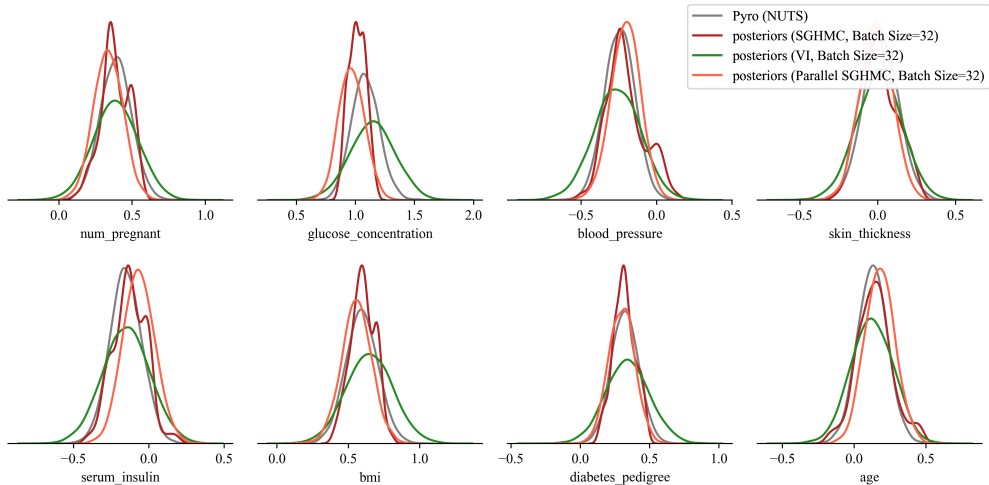

Figure 12: **Mini batch marginal distributions for Bayesian logistic regression**. Otherwise, same setup as Figure 11 with full batch Pyro retained as a baseline. Again all methods provide relatively consistent posterior approximations.

Table 3: **Metrics for Bayesian Logistic Regression.** Kernelized Stein discrepancy (KSD) Gorham and Mackey (2015) measures the distance between the samples provided by the algorithm and the true posterior via a kernel function (in this case a standard Gaussian). All results are averaged over 10 random seeds with one standard deviation displayed. [†]The displayed parallel SGHMC time represents the time for a single chain that could be obtained with sufficient parallel resources.

|  | KSD ↓ | Time (s) ↓ |
|---|---|---|
| Pyro (NUTS Hoffman et al. (2014)) | $2.680 \pm 0.001$ | $15.38 \pm 0.74$ |
| BlackJAX (SGHMC, Full Batch) | $2.689 \pm 0.013$ | $5.29 \pm 0.74$ |
| posteriors (VI, Full Batch) | $2.623 \pm 0.001$ | $7.04 \pm 0.23$ |
| posteriors (VI, Batch Size=32) | $2.516 \pm 0.003$ | $6.52 \pm 0.22$ |
| posteriors (SGHMC, Full Batch) | $2.693 \pm 0.024$ | $14.37 \pm 0.31$ |
| posteriors (SGHMC, Batch Size=32) | $2.686 \pm 0.012$ | $13.73 \pm 0.38$ |
| posteriors (Parallel SGHMC, Batch Size=32) | $2.690 \pm 0.001$ | $0.79 \pm 0.04^{\dagger}$ |

