# OpenReview forum: "Scalable Bayesian Learning with posteriors"
_ICLR.cc/2025/Conference — ICLR 2025 Poster_

### Official Review · Reviewer_zqMe · 2024-10-22

**Soundness:** 3
**Presentation:** 2
**Contribution:** 4
**Rating:** 6
**Confidence:** 4

**Summary:**

This paper proposes `posteriors`, a new PyTorch library for scalable Bayesian learning. Unlike previous libraries, `posteriors` is functional: The user can freely code their own minibatch-wise loss function and `posteriors` will use it to compute the posterior. Various Bayesian deep learning methods are available, such as the (linearized) Laplace approximation, variational inference, and stochastic-gradient MCMC.

The key selling points of `posteriors` is its composability, extensibility, swappablilty, and its functional nature. For instance, performing Bayesian inference on parameter-efficient fine-tuning in LLMs is a breeze since `posteriors` does not need to know the underlying model.

Moreover, the authors discussed a parallel deep ensemble technique and applied it to the Llama-3-8b model, effectively enabling this latest LLM model to detect out-of-distribution data. Further experiment results on inference techniques available in `posteriors`, such as the linearized Laplace for continual learning, were also presented in this paper.

**Strengths:**

I like the approach of `posteriors` in making Bayesian inference easier to perform, even in highly sophisticated models like LLMs. Indeed, by focusing in composability and by being functional-first means that `posteriors` is not restricted to a small class of models.

Moreover, I'm impressed by how clean and general the code of `posteriors` from the users' point-of-view (Fig. 3). I believe that `posteriors` is a useful addition to the community, enabling easy-to-do Bayesian inference on various models.

**Weaknesses:**

While the main contribution, i.e. the `posteriors` package, is great, the main weakness of this work is the presentation in the paper itself.

The story-telling of the paper is rather haphazard, badly motivated, and rather disjoint. The main purpose of this paper is to describe `posteriors`, to report the details of the package, and to show that it is useful. However, right from the introduction, the authors fail to motivate why do the community needs `posteriors` while other libraries such as `laplace-torch` exists, which also support sophisticated models like [LLMs](https://aleximmer.github.io/Laplace/huggingface_example/) and [reward models](https://aleximmer.github.io/Laplace/reward_modeling_example/). Indeed, the authors only motivate and compare `posteriors` with packages for traditional Bayesian inference (Line 86) and other functional libraries like Pyro in the Related Work section. Please note that I'm not saying that `posteriors` should not exist, but the motivation and comparison should be made stronger to make the paper more appealing to the readers, who are, ultimately, potential users.

In Section 3, I am underwhelmed by the fact that the authors did not describe their library in detail. As a library paper, I would expect the paper to be structured as in e.g. [PyTorch's paper](https://arxiv.org/pdf/1912.01703). I suggest the authors expand Section 3 more and reduce unnecessary discussion like Sec. 4, which ultimately does not contribute to the description of the library itself.

Indeed, Section 4 is rather disjoint from the rest of the paper. It is much more suitable for a method paper. If the main purpose, as the authors intended, is to showcase `posteriors`' extendability, then this section will be much better used to show _how_ to implement the method with `posteriors`, not to propose the method itself.

I am also confused about the selection of the experiments, especially Sec. 5.1. There, the authors only studied the cold posterior effects, which reads more like a validation of the parallel MCMC method proposed in Sec. 4. Again, it reads like a method paper, and thus disjointed from the rest of the paper.

Secs. 5.2 and 5.3 are fine in terms of the problems. But they would be much stronger if the authors showcased _how_ `posteriors` play a part in making this possible. For example, the code needed to construct such a Bayesian LLM is very short with `posteriors`. The failure to do so made the paper less coherent, and ultimately, it is hard for a reader to fully understand the benefits of `posteriors`.

Lastly, I noticed quite a bit of typo/misformatting in the paper, e.g. incorrect use of `\citep` and `\citet` (Line 157 etc), incorrect reference formatting like "A.1" instead of "Appendix A.1" in line 124.

In sum, I believe the paper needs several more writing iterations to make the story coherent, thus fully showcasing the potential of `posteriors`, which I do really like.

**Questions:**

The authors mentioned that `posteriors` is unable to provide model-aware techniques such as KFAC-Laplace. I believe that the future of Bayesian foundation models requires model-aware approaches: E.g. the current Laplace-LoRA with KFAC covariance (Yang et al., ICLR 2023) is insufficient since KFAC combined with LoRA has the same costs (at least memory-wise) as KFAC on the original weight matrices, thus negating the benefits of PEFT. (To see this, notice that KFAC on LoRA implies Kronecker factors of $n \times n$, $k \times k$, $k \times k$, and $m \times m$ for $n \times m$ original weights.) How does `posteriors` handles these limitations?

---

> ### Author Response · Authors · 2024-11-19
>
> > I like the approach of posteriors in making Bayesian inference easier to perform, even in highly sophisticated models like LLMs. Indeed, by focusing on composability and by being functional-first means that posteriors is not restricted to a small class of models.
> Moreover, I'm impressed by how clean and general the code of posteriors from the users' point-of-view (Fig. 3). I believe that posteriors is a useful addition to the community, enabling easy-to-do Bayesian inference on various models.
>
> We thank the reviewer for his very encouraging comments, and we also believe that composability, being mini-batch first, and functional are key features for large-scale Bayesian inference on modern neural network architectures.
>
> > the authors fail to motivate why do the community needs posteriors while other libraries such as laplace-torch exists
>
> The combination of highly desirable features described in Section 3 is to our knowledge unique to posteriors as well as being able to compose easily with the host of popular pretrained PyTorch models. We have added a comment after the bullet points in Section 3 (in addition to the related work in Section 6).
>
> > I would expect the paper to be structured as in e.g. PyTorch's paper. I suggest the authors expand Section 3 more and reduce unnecessary discussion like Sec. 4, which ultimately does not contribute to the description of the library itself.
>
> In structuring the paper, we took inspiration from other successful library papers such as [1] and [2] which have been published in venues like ICLR and have a general structure of (i) describe key mathematical components and motivations (ii) introduce library and behaviour (iii) experiments that are interesting in their own right (which are made accessible by the library). We feel `posteriors` more closely matches these applied libraries rather than a paper for a heavyweight library such as PyTorch (which goes into low-level code descriptions, including interfacing with C++ and tensor allocation). In the revision, we have expanded section 3 to include a new subsection 3.1 explicitly describing the methods included in `posteriors` and their complexities, therefore providing a more comprehensive description of the library.
>
> > I am also confused about the selection of the experiments
>
> In the introduction, we introduced our position on the three key benefits of Bayesian techniques in the large-scale deep learning regime. The experiments were therefore each selected to investigate one of these benefits and how they vary across approximation techniques, as made accessible through `posteriors`. We are encouraged that the other reviewers generally found the experiments convincing and appropriate for ICLR.
>
> > Lastly, I noticed quite a bit of typo/misformatting in the paper, e.g. incorrect use of \citep and \citet (Line 157 etc), incorrect reference formatting like "A.1" instead of "Appendix A.1" in line 124.
>
> Thank you for highlighting these typos. We have amended the revision.
>
> > The authors mentioned that posteriors is unable to provide model-aware techniques such as KFAC-Laplace
>
> K-FAC is not currently supported in `posteriors` as all current methods assume only an arbitrary `log_posterior` function and K-FAC requires awareness of the weight structure (further the original K-FAC approach [3] didn’t extend to architectures such as CNNs or transformers, although this has since been generalised to “weight-sharing” architectures [4]). Adding K-FAC is high priority on the `posteriors` roadmap but a key challenge is to formulate a general and modular K-FAC submodule that can be used for general models and across inference techniques (optimization, Laplace, VI). To our knowledge, such a functional implementation does not yet exist in the ecosystem. We have added extended discussion on K-FAC to the limitations section of the revision.
>
> Provided we have addressed your remarks, we would be really happy if you would consider increasing your rating.
>
> [1] E. Daxberger et al - Laplace Redux – Effortless Bayesian Deep Learning\
> [2] S. K. Choe - Betty: An Automatic Differentiation Library for Multilevel Optimization\
> [3] J. Martens and R. Grosse - Optimizing Neural Networks with Kronecker-factored Approximate Curvature\
> [4] R. Eschenhagen et al - Kronecker-Factored Approximate Curvature for Modern Neural Network Architectures

---

> ### Comment · Reviewer_zqMe · 2024-11-21
>
> Thanks for your rebuttal, and thanks for improving the text! Note that while I'm critical of the paper, this is out of a desire to see the paper to be successful if accepted since as I already mentioned, I really like the library itself.
>
> While the overall text has improved, I still think Sec. 4 and Sec. 5 need revisions. You mentioned that Daxberger et al., 2021 and Choe et al., 2023 are the inspiration when structuring the paper. Here are the problems:
> - I'd say Daxberger et al., 2021 is a review paper first and foremost, so it's important to show the general performance (i.e. not tied to the library) of the Laplace approximation. You could argue that your paper is also a review paper about Bayesian deep learning. But this weakens your case since such a review paper has been published---see Papamarkou et al.'s position paper at ICML 2024.
> - In the case of Choe et al., 2023, they're indeed closer to the present paper---it's a method+library paper. Note however that their presentation is quite different from your paper.
>     - Right in the intro Choe et al. presented the problem Betty is solving. I quote: _"[...] Nevertheless, research in gradient based MLO has been largely impeded by two major bottlenecks. First, implementing gradients in multilevel optimization, which is achieved by composing best-response Jacobians via the chain rule, requires both programming and mathematical proficiency. [...]"_  This is why in my original review, I encouraged you to clearly discuss "what kind of hole `posteriors` is trying to fill in", using `laplace-torch` as motivation.
>     - Choe et al. then presented their new method for efficient gradient computation and then presented Betty, which packages that method into a comprehensive library. In your paper, meanwhile, your main contribution is the package itself, so I advise you to really focus on `posteriors` itself. Choe et al., even if a hybrid method-library paper, clearly mentioned the implementation problem they're tackling in Sec. 4. Meanwhile, from reading your paper, it's hard to get what exactly the implementation problem `posteriors` is tackling. If it's minibatch-first & composability, then please expand them and give them the highlights.
>
> Then, my main issue with Sec. 4 still stands. Instead of a disjointed "Method" section out of nowhere, how about tying it to `posteriors` much more closely? For example, by really stressing that "due to the composability and minibatch-first design of `posteriors`, the following novel method can easily be implemented", then discuss the implementation detail extensively.
>
> Re. Sec. 5, it's a bit disjoint since, given that the paper is about the software library, the experiments are very general: More like a benchmark of Bayesian deep learning methods. I've already argued that this should be avoided. But you can salvage this section by following Choe et al. Notice in their paper that they show runtime and memory improvements due to their software in addition to performance numbers; basically saying to the reader "Hey, use Betty; see all these practical benefits!". Memory & runtime are indeed the main problems Choe et al. are solving. It's even stated right in the abstract. So, if your claim of `posteriors` is about easy-to-use and performance benefits, then show them prominently in the Experiment section.
>
> I hope this helps!

---

> > ### Author Response · Authors · 2024-11-25
> >
> > Thank you for your very constructive feedback, we are really appreciative of your thoughtful comments and active discussion to improve the paper.
> >
> > > I'd say Daxberger et al., 2021 is a review paper first and foremost, so it's important to show the general performance (i.e. not tied to the library) of the Laplace approximation. You could argue that your paper is also a review paper about Bayesian deep learning. But this weakens your case since such a review paper has been published---see Papamarkou et al.'s position paper at ICML 2024.
> >
> > Whilst of course subjective we don’t fully agree that Daxberger et al 2021 is a review paper first and foremost, for us their paper also puts prominent focus on the library and experiments (as stated clearly in their abstract) and so is comparable to the `posteriors` paper. Therefore we believe that for the `posteriors` paper it was still important to show general performance of Bayesian learning and justify the three advantages claimed in the introduction (this motivated the structure of the experiments). We also highlight that the review paper Papamarkou et al had no experiments and did not justify the general performance of Bayesian deep learning. However, we do agree that better highlighting `posteriors`’ features and utility improves the paper and we have amended the revision (see below).
> >
> > > Then, my main issue with Sec. 4 still stands. Instead of a disjointed "Method" section out of nowhere, how about tying it to posteriors much more closely? For example, by really stressing that "due to the composability and minibatch-first design of posteriors, the following novel method can easily be implemented", then discuss the implementation detail extensively.
> >
> > We have rewritten the start of Section 4 as well as parts of 4.1 and 4.2 to explain how the tempered viewpoint and SGMCMC as a generalization of stochastic optimization (and parallel-SGMCMC as a generalization of deep ensembles) guided the convenient “optimization-familiar” implementation (closely matching torchopt and optax) in posteriors to make Bayesian learning maximally accessible and scalable.
> >
> > > So, if your claim of posteriors is about easy-to-use and performance benefits, then show them prominently in the Experiment section.
> >
> > We have rewritten the start of Section 5 and added a paragraph at the start of each of the subsections describing how `posteriors` aided implementation in each experiment.
> >
> > Thanks again for your thoughtful comments and engaging in a productive discussion which has markedly improved the paper.
> >
> >
> > Daxberger, Erik, et al. "Laplace redux-effortless Bayesian deep learning."\
> > Papamarkou, Theodore, et al. "Position: Bayesian Deep Learning is Needed in the Age of Large-Scale AI."

---

> > > ### Comment · Reviewer_zqMe · 2024-11-25
> > >
> > > Thanks for the update! I think the text is much better now.
> > >
> > > We can argue for a long time on the exact nature of Daxberger et al. But I don't think this is productive. One thing for sure is that as a third-party reader, there is a noticeable difference when comparing Daxberger et al. (and Choe et al.) with the present paper, in that the formers are more coherent. Your paper, meanwhile, is trying to be so many things at once: library, review, and novel method. Again, this is not to shoot down your paper, but rather an honest take from me.
> > >
> > > I have adjusted my score accordingly.
> > >
> > > In any case, I highly suggest adding code examples to support your experiments in the appendix, refer to Appendix B of Choe et al. Basically, to showcase to the reader how easy it is to implement those methods.

---

### Official Review · Reviewer_pqmW · 2024-10-31

**Soundness:** 3
**Presentation:** 3
**Contribution:** 3
**Rating:** 8
**Confidence:** 3

**Summary:**

This paper introduces "posteriors", a PyTorch-based library designed to enhance scalability and usability of Bayesian inference for deep learning. The focus is on making Bayesian learning feasible for large datasets and models, often challenging due to the computational demands of approximating posterior distributions. Authors proposes a novel approach by reframing stochastic gradient MCMC as a form of gradient descent, creating a unified pathway between them. Experimental results highlight improvements in generalization, continual learning, and out-of-distribution detection, facilitated by scalable, minibatch-focused Bayesian techniques.

**Strengths:**

- the paper is mostly clear and concise, presents a straightforward approach, keeping primary discussions in the main sections and moving technical details to the appendix.
- the paper provides extensive experimentation, including large datasets and Bayesian inference with large-scale models, showing the library's potential across applications
- the library is composable, extendable, and scalable; and promotes compatibility with pytorch and other popular libraries, and could therefore have a relevant impact in the community
- the paper unifies stochastic gradient MCMC with traditional gradient descent, enabling efficient transitions between sampling and optimization
- the library is open-source, and provides a functional API that aligns with pytorc , and so facilitates user contributions

**Weaknesses:**

- the clarity of writing in some technical sections could be improved. Certain descriptions of key concepts, particularly the tempered SGMCMC framework (section 4), would benefit from a more step-by-step breakdown. I felt it was a bit too rushed.
- the main text could further emphasize how "posteriors" differentiates itself technically, showing specific implementations or optimizations unique to this library (in a similar spirit to Figure 3). The current presentation could be misinterpreted as implementing previously established methods rather than providing a novel platform for Bayesian inference
- although the experiments are robust, comparisons to other popular Bayesian libraries are quite limited. Explicitly comparing computational efficiency or accuracy with other Bayesian packages would better position "posteriors" in the current landscape.

**Questions:**

- could the authors clarify how "posteriors" manages scalability in large parameter regimes, particularly regarding memory management with large covariance matrices?
- how does "posteriors" compare in ease of use and computational efficiency with similar Bayesian packages like Pyro or BlackJAX?
- Given that parallel SGMCMC shows promising results in certain tasks, do the authors anticipate specific scenarios or types of models where it may not perform well, and if so, why?

---

> ### Author Response · Authors · 2024-11-19
>
> > could the authors clarify how "posteriors" manages scalability in large parameter regimes, particularly regarding memory management with large covariance matrices?
>
> All of the methods discussed in the paper have memory requirements linear in the number of parameters. Parameters can be stored by the user in arbitrary `TensorTree`s (including dicts, lists, tuples) and the `posteriors` Bayesian state will preserve this structure in a way that is transparent. As such, efficient memory and device management is mostly controlled by the user (in the same way as it would be with optimization in PyTorch). Specifically regarding covariance matrices, in the paper we have focussed on diagonal covariances however we plan to add low-rank and K-FAC support as a high priority, see response to reviewer zc1p. In the revision, we have added a new subsection 3.1 describing memory considerations and expanded on K-FAC in the limitations section.
>
> > how does "posteriors" compare in ease of use and computational efficiency with similar Bayesian packages like Pyro or BlackJAX?
>
> Unlike Pyro, `posteriors` is focussed on high dimensional models (such as neural networks) rather than traditional Bayesian probabilistic models (although they compose easily). The posteriors API more closely resembles an optimization setup (e.g. in torchopt or optax) rather than the `.run` or `.fit` methods utilized by Pyro or other PPLs. This allows the user more control over iterations and memory which is key for high dimensional inference (whilst still being user friendly, Figure 3). BlackJAX has a similar API but `posteriors` differentiates by being minibatch-first, increased unification and composing within the PyTorch ecosystem (thus enabling its integration with most HuggingFace models, for example).
>
> > Given that parallel SGMCMC shows promising results in certain tasks, do the authors anticipate specific scenarios or types of models where it may not perform well, and if so, why?
>
> This is a very exciting area of future research for us! As introduced in the paper, parallel SGMCMC is closely connected to deep ensembles and therefore generally shares strengths and limitations. As with deep ensembles, performance is expected to be strong across model types (see e.g. [1]) and limitations are generally resource-centric since parallel SGMCMC has large memory and parallelisation requirements. In the revision, we’ve added discussion on the limitations of parallel SGMCMC at the end of Section 4.
>
> [1] J. Z. Liu et al - Simple and Principled Uncertainty Estimation with Deterministic Deep Learning via Distance Awareness

---

> > ### Comment · Reviewer_pqmW · 2024-11-26
> >
> > I thank the authors for the clear answer. I keep supporting the paper for acceptance.

---

### Official Review · Reviewer_cMYJ · 2024-11-03

**Soundness:** 3
**Presentation:** 2
**Contribution:** 3
**Rating:** 6
**Confidence:** 4

**Summary:**

I have increased my score from 5 to 6.
-----
This paper presents a pytorch library called `posteriors` for general-purpose Bayesian learning tasks, which is featured by (1) composability, (2) extensibility, (3) functionality, (4) scalability, and (5) swappability. The implementation includes a tempered version of Parallel SGMCMC, which is shown to combine the favorable properties of deep ensembles and serial SGMCMC. Empirically, the library has been tested for cold posteriors, continual learning with LORA, and finetuning of Llamma3 with Bayesian learning, demonstrating the benefits of Bayesian paradigm for machine learning.

**Strengths:**

The paper is in general well-written and the empirical results are presented clearly. I like the key features of the proposed library which are lacking in existing libraries. Below are the key strengths.

- The key features of the proposed library as sketched in section 3 are important for modern Bayesian deep learning.

- The experiments cover various scenarios in current deep learning landscape.

- the library is open-sourced and supports customization.

**Weaknesses:**

I think the paper can be improved by better scoping the problems that the library aims to solve, the inference approaches provided by the library, and the practical tradeoffs between the proposed Bayesian approach versus the non-Bayesian approaches (e.g. SGD, LoRA and Deep Ensemles and their various combinations).

In particular, the following questions should be addressed:


- What problems can the proposed library solve? Examples include disentangling various sources of uncertainties, continualing learning, finetuning, standard classification / regression,  classical Bayesian inference problems e.g. time series, hierarchical modeling? Detailing these problems will help readers understand what set this approach different from the existing ones such as Stan and BlackJAX. And perhaps a table that outlines the differences among these approaches can be helpful.

- What are the key inference algorithms currently supported in the library?

- What is the practical time and memory complexity of the current implementations when compared against the existing non-Bayesian approaches?

-- What are the limitations / scenarios that the current library does not support?

Overall I think the proposed library is very promising. And the throughout the paper some of the above points have been touched upon briefly. However, a comprehensive discussion is lacking, which is critical to the potential target users of the library. I will consider raising my scores if the above points are addressed during rebuttal.

**Questions:**

- figure 4: only 4 lines are plotted but 5 lines are included in the legend box?

---

> ### Author Response · Authors · 2024-11-19
>
> > What problems can the proposed library solve?
>
> In general, the value of Bayesian methods is widely recognized (e.g. [7]), however their computational cost has limited widespread adoption, the goal of `posteriors` is to help remove these barriers. Establishing the benefits of Bayesian inference is an active area of research and indeed varies across fields and applications. In the introduction and Figure 1 we have described and taken a position on the three key problems (generalization, continual learning, OOD detection) that Bayesian inference solves in the context of large-scale deep learning, which aligns with [6]. However, as you highlight in other fields such as more interpretable models there are additional benefits (e.g. [1]) which we have commented on in the introduction in the revision.
>
> > Detailing these problems will help readers understand what set this approach different from the existing ones such as Stan and BlackJAX. And perhaps a table that outlines the differences among these approaches can be helpful.
>
> We have highlighted the key unwavering design principles of `posteriors` in Section 3 and a description of alternative packages.  Important differentiators of `posteriors` are that it is minibatch-first and functional PyTorch based allowing easy use with e.g. the majority of HuggingFace models.
>
> > What are the key inference algorithms currently supported in the library?
>
>
> The key Bayesian inference algorithms are SGMCMC, VI and Laplace. Additionally, extended Kalman methods [2] are supported as well as optimization via torchopt within the `posteriors` unified API. The `posteriors` inference framework is extremely general and we will be able to easily add algorithms in the future including SVGD [3], pre-conditioned SGMCMC [4] and more. We have added a new subsection 3.1 to the revision describing included inference algorithms.
>
> > What is the practical time and memory complexity of the current implementations when compared against the existing non-Bayesian approaches?
>
> All approaches have memory that is linear in the dimension of the parameters (aside from the newly added dense Laplace and VI methods), however methods that formulate a Fisher information matrix (Laplace, extended Kalman) require O(batchsize * d) memory which can be a bottleneck for very large models (although the batch size can be reduced). Time complexity is more subtle since the different algorithms have different convergence criteria. We’ve added a discussion on time and memory complexity of supported methods to the new subsection 3.1.
>
> > What are the limitations / scenarios that the current library does not support?
>
> We’ve described in subsection 3.2 (previously 3.1) the limitations of the library, in particular the general `log_posterior` formulation makes it prohibitive to utilize loss function specific approximations. `posteriors` also doesn’t have its own PPL-style distribution constructors in favour of composition with Pyro. Are there some specific scenarios you had in mind that weren’t clear whether `posteriors` supported? We would be very interested to add a discussion of such scenarios in the revised version of the manuscript.
>
> > figure 4: only 4 lines are plotted but 5 lines are included in the legend box?
>
> Non-linearized Laplace EF and Laplace GGN overlap here, hence we’ve added a comment clarifying to the caption. This is due to the Laplace approximation in the high dimensional regime performing badly without linearization which agrees with the literature, see e.g. [5].
>
> [1] A. Gelman et al - Bayesian Workflow\
> [2] Y. Ollivier - Online Natural Gradient as a Kalman Filter\
> [3] Q. Liu and D. Wang - Stein Variational Gradient Descent: A General Purpose Bayesian Inference Algorithm\
> [4] Y. A. Ma et al - A Complete Recipe for Stochastic Gradient MCMC\
> [5] A. X. Yang et al - Bayesian Low-rank Adaptation for Large Language Models\
> [6] T. Papamarkou - Position: Bayesian Deep Learning is Needed in the Age of Large-Scale AI\
> [7] S. Kapoor - Large Language Models Must Be Taught to Know What They Don't Know

---

> > ### Comment · Reviewer_cMYJ · 2024-11-26
> >
> > Thank you for the rebuttal. The added discussion has addressed most of my concerns.  I will increase my score to 6.

---

### Official Review · Reviewer_zc1P · 2024-11-03

**Soundness:** 3
**Presentation:** 3
**Contribution:** 3
**Rating:** 8
**Confidence:** 4

**Summary:**

The paper introduces the `posteriors` pytorch library for programmatic approximate Bayesian inference with a focus on deep neural networks as models, which further contains some advances, like a variant of stochastic gradient Markov chain Monte Carlo. The paper assesses the implemented methods on a variety of problems.

**Strengths:**

- Software release that allows other researchers and practitioners to build on the proposed and baseline methods. Overall appears to contain MCMC-based methods, variational inference, and Laplace approximations, all of which are relevant and practical methods.
- Clearly written motivation for Bayesian inference in general
- Interface seems very clear and flexible due to functional paradigm
- Additional contribution in form of a new SGMCMC algorithm?
- Convincing experiments on modern architectures like lora and llama.

**Weaknesses:**

- Only diagonal variational and Laplace approximations, which are known to have certain issues and are generally outperformed by more structured posterior approximations. For example, Laplace approximations perform better across the board when using layer-wise Kronecker-factored structure (Ritter et al, ICLR 2018, https://discovery.ucl.ac.uk/id/eprint/10080902/1/kflaplace.pdf). Apparently, extending the library to such approximations seems out of scope? (lines 209-210)
- I find the interweaved SGMCMC contribution hard to understand and it appears to break the story a bit. Maybe it misses parts of the appendix to be understandable by itself.

**Questions:**

- Would it be possible to extend the library to non-diagonal parametric posterior approximations in the future?
- Could the authors clarify their SGMCMC contribution in relation to recent prior work on such algorithms?
- How were priors chosen in the experiments on the cold posterior effect? For VI and Laplace, these can have quite a strong effect and I couldn't find any information on that.
- How would Figure 7 look like for the MAP as a comparison? I have seen claims that large models are already fairly well calibrated when looking at confidence itself.

---

> ### Author Response · Authors · 2024-11-19
>
> > Would it be possible to extend the library to non-diagonal parametric posterior approximations in the future?
>
> Since the paper submission dense Laplace and VI methods have been added by new contributors alongside broader community building. Naturally, low-rank and K-FAC approximations are more scalable and represent a high priority as additional features for the library and certainly not out of scope. The main issue with K-FAC is that the original approach [1] was limited to feedforward networks (mat-mul followed by elementwise non-linearity) and not immediately transferable to general/modern architectures (CNNs, transformers). More recent work [2] has generalised this to a “weight-sharing” framework, yet it still remains a challenge for us as `posteriors` developers to establish a convenient, functional API to construct K-FAC abstractions that can be used across inference algorithms (optimization, Laplace, VI) and within the `log_posterior` framework. We have expanded our discussion on this in the limitations section of the revised paper.
>
> > Could the authors clarify their SGMCMC contribution in relation to recent prior work on such algorithms?
>
>
> `posteriors` does not contain any new SGMCMC algorithms (although it has been designed so that they can be easily added in the future). However, our implementation enables a  temperature parameter that allows a seamless transition between SGMCMC and stochastic gradient optimization (via temperature=0). This is not altogether novel, (e.g. discussed in Appendix C of [3]) though we haven’t seen it discussed in the general SGMCMC framework of [4]. Additionally, we present a connection between deep ensembles and parallel MCMC. This is is novel (to the best of our knowledge). We have clarified this at the start of Section 4 in the revision.
>
> > How were priors chosen in the experiments on the cold posterior effect? For VI and Laplace, these can have quite a strong effect and I couldn't find any information on that.
>
> The effect of the prior varies depending on the problem. Due to length limitations, we decided not to do an in-depth study into prior specification in this paper. For the cold posterior effect, we have followed exactly [5] and used a fixed diagonal Gaussian prior with all variances set to 1/40, where in [5] they show Bayesian predictions to be robust to the choice of prior scale for this specific problem.
>
> > How would Figure 7 look like for the MAP as a comparison? I have seen claims that large models are already fairly well calibrated when looking at confidence itself.
>
> Although large models can have good calibration in more simple classification tasks, we don’t in general expect this to be the case in comparison to Bayesian approaches (e.g. [7]). In particular, access to epistemic uncertainty (separate from semantic uncertainty) is key for open-ended natural language generation (see [6] for more details). Specifically for Figure 7, the first pane is the base Llama 3 model and can therefore be considered the MLE on just the Llama training data, the second pane is the MLE after finetuning. In both cases, we wouldn’t expect or indeed observe the non-Bayesian MLE (or MAP) techniques to be able to accurately differentiate epistemic uncertainty from semantic uncertainty via the single total entropy metric. We have clarified this in Section 5.3 of the revision.
>
> [1] J. Martens and R. Grosse - Optimizing Neural Networks with Kronecker-factored Approximate Curvature\
> [2] R. Eschenhagen et al - Kronecker-Factored Approximate Curvature for Modern Neural Network Architectures\
> [3] F. Wenzel et al - How good is the Bayes posterior in deep neural networks really?\
> [4] Y. A. Ma et al - A Complete Recipe for Stochastic Gradient MCMC\
> [5] P. Izmailov et al - What are Bayesian neural network posteriors really like?\
> [6] L. Kuhn et al - Semantic Uncertainty: Linguistic Invariances for Uncertainty Estimation in Natural Language Generation\
> [7] J. Z. Liu et al - Simple and Principled Uncertainty Estimation with Deterministic Deep Learning via Distance Awareness

---

> ### Comment · Reviewer_zc1P · 2024-11-26
>
> Thank you for the clarifications and updated text in the paper. I remain optimistic about the paper and `posteriors` package and therefore keep my score.

---

### Author Response · Authors · 2024-11-19
**Summary of Revision**

We thank the reviewers for their thorough and thoughtful feedback on our submission. We are grateful that the reviewers appreciated the contribution made through posteriors, namely to provide a scalable Bayesian inference library that is mini-batch first and compatible with PyTorch (and is the first of this kind). We are also grateful for valuable comments on the presentation which has improved the quality and clarity of the revision.

In light of reviewer comments, we have made the following changes to the attached revised manuscript:

- Expanded the first paragraph in the introduction to clarify additional benefits for classical statistical models (beyond the three key features highlighted in the large scale ML regime) as well as early highlighting of the motivation for the posteriors package.
- Added a new Section 3.1 describing the methods currently supported in the posteriors package and their complexities.
- Added a paragraph to Section 3.2 (Limitations) dedicated to K-FAC and how its use of model structure means it is not currently supported by posteriors’ however support has been scoped out.
- Expanded the start of Section 4 to highlight the novelty in the tempered connection between optimization and SGMCMC as well as the link to deep ensembles.
- Discussion at the end of Section 4 detailing limitations of parallel SGMCMC.
- Extended discussion in Section 5.3 on improved out-of-distribution detection capabilities of Bayesian methods over point-estimates.

All changes are highlighted in blue in the revised manuscript.

We are open to further revise the presentation aspects of the paper during the discussion period.

UPDATE: After productive discussion with Reviewer zqMe, we have additionally revised the manuscript with the following points
- We have rewritten the start of Section 4 as well as parts of 4.1 and 4.2 to better tie in the tempered unification of SGD, SGMCMC and deep ensembles with the implementation and features of the `posteriors` library.
- We have added a paragraph to the start of each of the subsections in Section 5 to highlight how the design of `posteriors` facilitated easy implementation for each of the experiments.

---

### Meta-Review · Area_Chair_4RKS · 2024-12-21

**Metareview:**

This paper introduces a new PyTorch library, `posteriors,` for approximate Bayesian inference with deep neural networks and foundation models. Unlike existing frameworks like Pyro or BlackJAX, this library focuses on Bayesian inference on deep models with a prescriptive list of possible inference algorithms (SGMCMC, VI, and Laplace). The authors include experiments that demonstrate models that can be easily implemented with their library.

The library has some limitations, notably that it does not allow for non-diagonal covariance approximations. This omission limits the package's applicability, though the advantage is that all other methods are linear time/memory. Moreover, the presentation of this paper could be improved to motivate the package better, describe limitations, and provide usable code examples for potential users.

Nevertheless, this package will be of interest to the community. It will increase the adoption of Bayesian inference methods, mainly when used with parameter-efficient fine-tuning and LLMs. Given that the authors made a significant effort during the revision to improve the manuscript, I recommend accepting this paper. However, I strongly encourage the authors to continue to improve the manuscript before the camera-ready deadline to address reviewers' suggestions - in particular with the inclusion of code examples - to best facilitate their library's interest and adoption.

**Additional Comments On Reviewer Discussion:**

Many reviewers raised questions about how this package differs from Pyro, BlackJAX, and other probabilistic programming languages. The authors noted the focus on deep models and updated the manuscript to reflect this.

Many reviewers also noted the lack of full-covariance approximate inference methods. The authors gave a reason for not including them, and reviewers suggested that a limitations section be included.

Reviewers raised some minor technical issues (e.g., how priors were chosen for cold posterior experiments), which the authors adequately addressed.

Many reviewers shared the most relevant comments regarding the paper's presentation. One reviewer, in particular, noted that this paper might be more effective if it were structured like successful software papers, such as the PyTorch paper. The authors made steps towards implementing some of these suggestions during the revision period, though there are still outstanding suggestions that could improve the paper.

Ultimately, all reviewers found that this package had the potential to benefit the community and thus make a significant contribution. Because the authors made great strides toward ameliorating concerns about presentation, I was inclined to accept the paper.

---

### Decision · Program_Chairs · 2025-01-22

Accept (Poster)